neuroscience

beat perception, auditory cortex, rhythm, electrophysiology, entrainment, temporal processing

**Authors for correspondence:**
Vani G. Rajendran
e-mail: vrajendr@cityu.edu.hk
Jan W. H. Schnupp
e-mail: wschnupp@cityu.edu.hk

# Auditory cortical representation of music favours the perceived beat

Vani G. Rajendran[1,2], Nicol S. Harper[1]
and Jan W. H. Schnupp[1,2]

[1]Auditory Neuroscience Group, Department of Physiology, Anatomy, and Genetics, University of Oxford, Oxford, UK
[2]Department of Biomedical Sciences, City University of Hong Kong, Kowloon Tong, Hong Kong

 VGR, 0000-0002-9734-9289; NSH, 0000-0002-7851-4840;
JWHS, 0000-0002-2604-0057

Previous research has shown that musical beat perception is a surprisingly complex phenomenon involving widespread neural coordination across higher-order sensory, motor and cognitive areas. However, the question of how low-level auditory processing must necessarily shape these dynamics, and therefore perception, is not well understood. Here, we present evidence that the auditory cortical representation of music, even in the absence of motor or top-down activations, already favours the beat that will be perceived. Extracellular firing rates in the rat auditory cortex were recorded in response to 20 musical excerpts diverse in tempo and genre, for which musical beat perception had been characterized by the tapping behaviour of 40 human listeners. We found that firing rates in the rat auditory cortex were on average higher on the beat than off the beat. This 'neural emphasis' distinguished the beat that was perceived from other possible interpretations of the beat, was predictive of the degree of tapping consensus across human listeners, and was accounted for by a spectrotemporal receptive field model. These findings strongly suggest that the 'bottom-up' processing of music performed by the auditory system predisposes the timing and clarity of the perceived musical beat.

## 1. Introduction

The perception of a steady pulse or beat in music arises from the interaction between rhythmic sounds and the way our brains process them. Two things make musical beat perception particularly intriguing. Firstly, no primate species apart from humans consistently shows spontaneous motor entrainment to the beat in music (e.g. tapping a foot, nodding the head) [1–4]. Secondly, despite musical beats being a subjective percept rather than an acoustic feature of music [5], individual listeners tend to

overwhelmingly agree on when beats occur. Some of this consistency might be due to certain 'top-down' constraints such as cultural and cognitive priors [6–8]. However, little is known about the contribution of 'bottom-up' auditory processing to the emergence of musical beat.

Recent cross-species work, which takes advantage of the fact that the auditory system is highly conserved across mammalian species [9–12], has shown that the beat that human listeners tap while listening to non-isochronous rhythms is accompanied by higher firing rates in the auditory midbrain on the beat than off the beat [13]. Importantly, this effect was entirely explained by neuronal adaptation, which differentially encodes otherwise identical sounds that differ only in temporal context. If points of relative neural 'emphasis' in the ascending auditory representation of rhythms predispose the induction of musical beat, then this could help explain why people largely agree on when beats occur. A key assumption here is that localized, transient increases in firing rates of auditory neurons would lead to points of perceptual emphasis, and that the temporal structure of these points in turn shapes the perception of a periodic beat. The idea that perceptual emphasis and beat perception are likely to be intimately linked is not new. Pioneering work by Paul Fraisse explored how differences in a sound feature (e.g. intensity, pitch) in a series of isochronous sounds immediately evoke the perception of rhythmic groupings (see e.g. [14] for a review), and Povel & Essens' [15] empirical model of beat perception suggests that the beat aligns itself maximally to 'perceptual accents' resulting from changes in a sound feature or temporal context. Nowadays, the cortical activity evoked by music in the human brain is thought to arise from interactions between relatively low-level evoked responses in the incoming sensory stream and 'higher-level' or feedback mechanisms that may include the entrainment of cortical oscillations [16]. Thus, a clearer understanding of the bottom-up neural signals evoked by music could shed light on how oscillatory dynamics in the brain entrain to auditory rhythms [17–22], as well as on the role played by the motor system in finding and maintaining a regular pulse [23–33]. By tracing the neural representation of rhythmic sounds through the nervous system and identifying points of divergence between humans and animal models, we may also begin to understand the origins and species-specificity of beat perception and synchronization [34].

However, the hypothesis that auditory processing creates points of neural emphasis that shape the perception of musical beat must first pass a crucial test; it must hold true not just for simple 'laboratory' rhythms, but also for real music. The aim of this study was to critically test whether transient increases in firing rate (neural 'accents') due to bottom-up auditory processing could explain when beats are felt in real recordings of music. Here, we show in the context of real music that firing rates are on average higher on the beat than off the beat. Furthermore, the extent of the asymmetry between on-beat and off-beat firing rates (termed 'beat contrast') predicts not only when the beat is likely to be perceived, but also how closely listeners agree with each other on when beats occur. To assess whether these effects are due to central auditory processing, we also computed responses of an auditory nerve model to the same musical excerpts, and found that the neural contrasts between on-beat and off-beat responses are less pronounced and less selective for the perceived musical beat structures compared to auditory cortical responses. Finally, we demonstrate that the beat-related signals expressed in cortical responses can be explained by the neurons' spectrotemporal receptive fields. These findings add to growing evidence that musical beat perception may be highly constrained by the low-level temporal and spectrotemporal contrast sensitivity of neurons in the ascending auditory pathway.

# 2. Methods

All stimuli, data and code for this study are available from the Dryad Digital Repository [35].

## 2.1. Stimuli

Twenty musical excerpts [5], which were diverse in tempo and musical genre (see electronic supplementary material, table S1), were played to three anaesthetized rats while recording extracellularly from auditory cortex. Recorded firing rates were then compared to tapping data collected from 40 human participants listening to the same musical excerpts. The 20 musical excerpts tested comprised the training dataset for the MIREX 2006 beat tracking algorithm competition [36]. Included in this published dataset were beat annotations collected from 40 human listeners tapping to

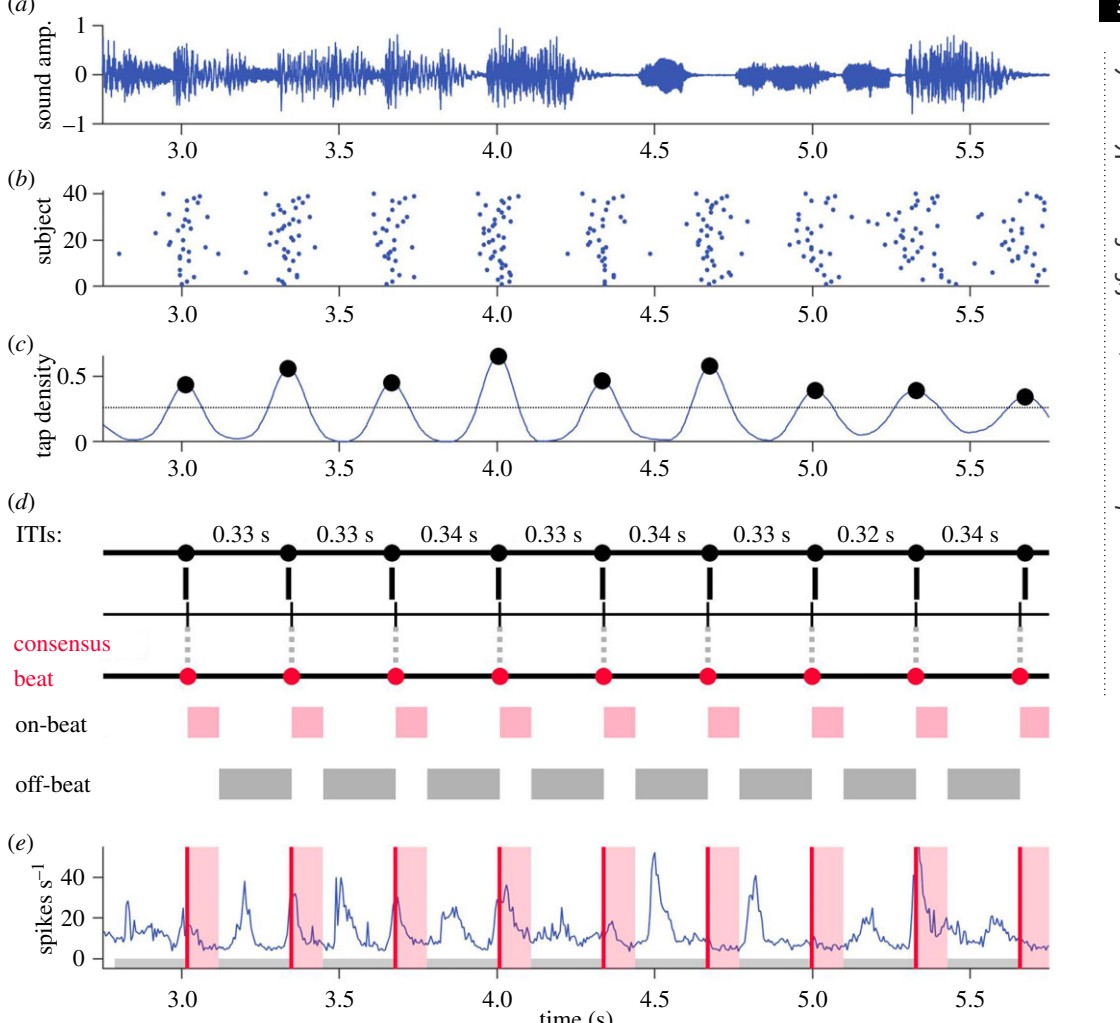

**Figure 1.** Finding the consensus beat. (*a*) Short excerpt from an example song. (*b*) Raster plot of corresponding human tap times. Rows are subjects, each dot represents a tap. (*c*) Smoothed pooled tap histogram, dots mark peaks found by peak-finder. (*d*) Inter-tap intervals (ITIs) between neighbouring peaks are not perfectly identical. The median of these values was taken as a song's consensus ITI. Aligning a grid with consensus ITI spacing (second line) to minimize the error between itself and the peaks in panel *c* results in consensus beat times (red dots), which have a constant beat period and phase. On-beat was defined as the 100 ms following consensus beats (red), and off-beat is all time excluding these windows (grey). (*e*) Population firing rate in the rat auditory cortex during the illustrated excerpt. Consensus beat times (red lines), on-beat windows (red shading) and off-beat windows (grey) are overlaid.

these musical excerpts [5]. Only the first 10 s of musical excerpts and beat annotations were used in this study, and all listeners began tapping a steady beat well within this time.

## 2.2. Tapping analysis

### 2.2.1. Consensus beat

Tap annotations from the 40 human subjects were combined to determine a single 'consensus' beat interpretation for each song. Figure 1 illustrates this procedure. To calculate consensus tap locations, tap times across the 40 subjects were pooled, binned into 2 ms bins and then smoothed using a Gaussian kernel with a width (standard deviation) of 40 ms (figure 1*c*). The precise width of the smoothing kernel was not critical to the results as long as it roughly matched the spread in the data. A peak-finder (Matlab function *findpeaks.m*) was then used to identify peaks that were larger than 40% of the maximum value in the smoothed histogram. The consensus tempo for each song was taken to

be the median interval between successive peaks. The consensus phase was determined by finding the offset that minimized the error between a temporal grid with consensus tempo spacing and the peaks found by the peak-finder (figure 1*d*). Consensus tap times for each song thus had a constant inter-tap interval (ITI; beat period) and offset (beat phase). Consensus beat rates ranged from 0.7 to 3.7 Hz, or 42 to 222 beats per minute, corresponding to beat periods of 1.42 down to 0.27 s for the 20 musical excerpts (see electronic supplementary material, table S1).

### 2.2.2. Strength of tapping consensus

The strength of the tapping consensus for each song was computed as the correlation between the observed tap distribution histogram (pooled across all listeners) and the 'ideal' tap histogram that would have resulted if all 40 listeners had interpreted the beat identically. The ideal case therefore assumes that all listeners tap precisely at each consensus beat time, and that any variation in tap times would only reflect motor error. An idealized tap histogram was constructed for each song by convolving consensus tap times with a Gaussian kernel whose width was 5% of the consensus beat period to add motor error that is consistent with the magnitude of errors reported in studies of human sensorimotor synchronization [37–39]. Other kernel widths close to 5% also produced consistent results. Next, both the pooled histogram of observed taps and the idealized tap histogram with added motor error were convolved with a 5% Gaussian kernel to perform kernel density estimation, which estimates the probability density function of tapping over time. The correlation coefficient between observed and idealized tap probability density functions was taken as a measure of the strength of the tapping consensus for each musical excerpt, where a large value would indicate close agreement between real tapping behaviour and 'ideal' behaviour that is free of perceptual uncertainty. We chose this particular correlation measure (instead of e.g. simply quantifying the variance of observed tap times around consensus beats) in order to avoid excessively penalizing minority but metre-related beat interpretations that are reported with high precision, but at a multiple of the speed of the majority of listeners (see electronic supplementary material, figure S5, excerpt 10, for example).

## 2.3. Surgical protocol

All procedures were approved and licensed by the UK home office in accordance with governing legislation (ASPA 1986). Three female Lister hooded rats weighing approximately 250 g were anaesthetized with 0.05 ml domitor and 0.1 ml ketamine i.p. To maintain anaesthesia, a saline solution containing 16 µg kg$^{-1}$ h$^{-1}$ domitor, 4 mg kg$^{-1}$ h$^{-1}$ ketamine and 0.5 mg kg$^{-1}$ h$^{-1}$ torbugesic was infused continuously during recording at a rate of 1 ml h$^{-1}$. A craniotomy was performed centred at 4.7 mm caudal to bregma and 3.5 mm lateral to the midline on the right-hand side.

Recordings were made using a 64 channel silicon probe (Neuronexus Technologies, Ann Arbor, MI, USA) with 175 µm$^2$ recording sites arranged in a square grid pattern at 0.2 mm intervals along eight shanks with eight channels per shank. The probe was first positioned over the auditory cortex based on anatomical coordinates and then inserted into the brain in a medio-lateral orientation if possible until all channels were inside the brain. After a few minutes, a search stimulus consisting of broadband noise bursts was played to check that recording sites were driven by sound. Next, frequency response areas (FRAs) were measured to check that channels were frequency tuned, and then the music stimuli were presented. Stimuli were presented binaurally through headphones at roughly 80 dB SPL at a sampling rate of 48828.125 Hz, and data were acquired at 24414.0625 Hz using a TDT system 3 recording set-up (Tucker Davis Technologies). The 20 musical excerpts were played in randomized order for a total of 12 repeats, with 3 s of silence separating songs.

## 2.4. Electrophysiology analysis

### 2.4.1. Data preprocessing

Offline spike sorting was performed using the expectation–maximization algorithm 'Klustakwik' [40] with manual post-processing using 'Klustaviewa' (Cortical Processing Lab, University College London). Each resulting cluster of spikes, which putatively originates from a small population of neurons near a recording site, is termed a multiunit. Firing rates over time for multiunits were

calculated by binning spike times into 5 ms bins, which resulted in peri-stimulus time histograms (PSTHs) at an effective sampling rate of 200 Hz.

Only multiunits that were reliably stimulus-driven were included in further analysis. To quantify the reliability of firing, we used a noise power to signal power metric developed by Sahani & Linden [41], which examines how repeatable neural response patterns are for repeat presentations of the same stimulus. Neural responses to the first 1 s of all 20 excerpts were concatenated together for this analysis. In line with established practice [42], multiunits that failed to show a noise power to signal power ratio less than 40 based on the 12 repeats were excluded from further analysis, leaving a total of 77 multiunits. All subsequent analyses were performed using custom-written Matlab code.

### 2.4.2. On-beat, off-beat and beat contrast

On-beat neural activity was defined as the average population firing rate in the 100 ms following consensus tap times, and off-beat neural activity was the average population firing rate during all time excluding on-beat windows (figure 1e). The justification for this definition is that (i) the true perceived beat location is probably just after a listener taps given that listeners tend to tap several tens of milliseconds earlier than the beat (negative mean asynchrony) [43], (ii) defining off-beat activity as all neural activity that is not on the beat is consistent with previous work [13], and (iii) a window of 100 ms is less than half a beat cycle for the fastest beat period in these data of 273 ms. The precise choice of time window is not critical, and this was confirmed by running all analyses using on-beat windows that ranged between 80 and 120 ms in 10 ms increments. The results were consistent with those presented here for a time window of 100 ms, and if anything, slightly stronger for shorter time windows.

To quantify the asymmetry between on-beat and off-beat responses, we adapted the Michelson contrast measure commonly used in vision research [44], and defined 'beat contrast' (BC) for a set of neural responses as:

$$BC = \frac{f_{on} - f_{off}}{f_{on} + f_{off}},$$

where $f_{on}$ and $f_{off}$ are mean on-beat and mean off-beat firing rates, respectively. Beat contrast would thus be 1 if $f_{on} \gg f_{off}$, −1 if $f_{off} \gg f_{on}$ and 0 if $f_{on} = f_{off}$.

### 2.4.3. Fitting the linear–nonlinear model

The relevant scripts for fitting spectrotemporal receptive field (STRF) and linear–nonlinear (LN) models are available on Github [45]. First, music stimuli were transformed into a simple approximation of the activity pattern received by the auditory pathway by calculating the log-scaled spectrogram ('cochleagram') [46,47]. For each sound, the power spectrogram was taken using 10 ms Hanning windows, overlapping by 5 ms. The power across neighbouring Fourier frequency components was then aggregated using overlapping triangular windows comprising 30 frequency channels with centre frequencies ranging from 150 to 22 833 Hz (1/4 octave spacing). Next, the log was taken of the power in each time-frequency bin, and finally any values below a low threshold were set to that threshold. These calculations were performed using code adapted from melbank.m (http://www.ee.ic.ac.uk/hp/staff/dmb/voicebox/voicebox.html).

The LN-STRF model was trained to predict the firing rate at time $t$ from a snippet of the cochleagram extending 100 ms (20 time bins) back in time from time $t$, using methods described in detail in Willmore et al. [46]. Briefly, the linear weights describing the firing rate of each neuron were estimated by regressing, with elastic net regularization, each neuron's firing rate at each time point against the 100 ms cochleagram snippet directly preceding it. A sigmoidal nonlinearity [42] was then fitted to constrain the range of firing rates and map from the linear activation to the predicted PSTH such that it minimized the error between the predicted PSTH and the observed PSTH. Each multiunit's best frequency (BF) was determined by finding the frequency band in the linear kernel with the largest positive weight. LN model predictions of a multiunit's PSTH to a test song were made by first convolving the cochleagram of the test song with the linear STRF and then applying the nonlinearity. Each multiunit's LN model was calculated 20 times, each time setting aside a different song as the test set, and PSTH predictions for each excerpt were made using the LN model that was not trained on that excerpt. Linear STRFs for all multiunits are shown in electronic supplementary material, figure S7, averaged across the 20 model runs.

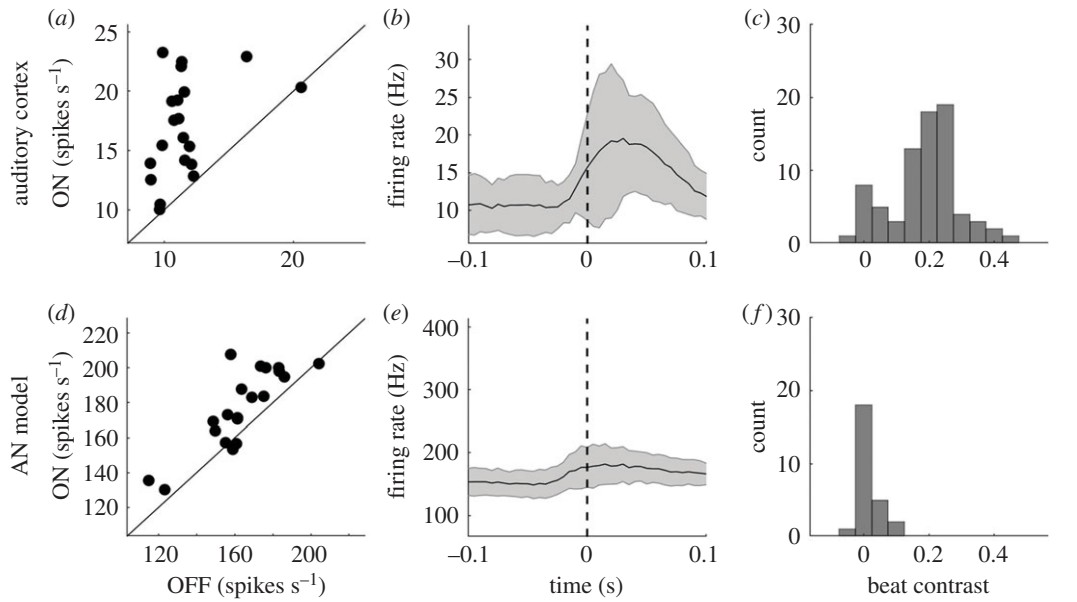

**Figure 2.** Neural activity is stronger on the beat than off the beat. (*a*) Mean on-beat versus off-beat population firing rate in auditory cortical neurons, dots are musical excerpts. On-beat firing rates are significantly higher than off-beat firing rates. (*b*) Grand average population firing rate in the auditory cortex in a 200 ms window around all consensus beats in all 20 excerpts, ± standard deviation. (*c*) Histogram of beat contrast (BC) values for each recorded multiunit (*N* = 77). (*d–f*) Same as panels *a–c*, but for population activity based on AN model responses (*N* = 26 model AN fibres). AN model firing rates are also higher on the beat than off the beat, but its corresponding BCs are much smaller.

### 2.4.4. Comparison to auditory nerve model responses

We also wanted to examine whether any observed beat contrast (BC) in the neural responses might be the result of peripheral, cochlear, rather than central auditory mechanisms. To this end, we used the well-established model of the auditory periphery by Zilany *et al.* [48] to model the responses of a set of auditory nerve (AN) fibres to each of the 20 musical excerpts. We then computed BC values for the simulated AN responses in a manner entirely analogous to that used for the cortical data. In order to make the comparison as valid as possible, we modelled a set of low spontaneous rate AN fibres, as saturation of responses in high spontaneous rate fibres might reduce BC. Additionally, we matched the range of model AN fibre characteristic frequencies (CFs) to that observed in our cortical multiunit best frequencies determined by STRF analysis, resulting in 26 model AN fibres with CFs between 300 and 22 833 Hz in quarter octave steps.

# 3. Results

## 3.1. Auditory cortical firing rates are higher on the beat than off the beat

Figure 2*a* shows the average on-beat population firing rate, averaged across all recordings from all rats, plotted against the off-beat population firing rate for each of the 20 tested musical excerpts. Firing rates on the beat were significantly larger than firing rates off the beat ($p < 10^{-4}$, Wilcoxon paired signed-rank test, $N = 20$ songs), an observation that is consistent with previous work examining auditory midbrain responses to simple rhythmic patterns [13]. The average population firing rate in the 200 ms window around consensus beats (averaged across all beats in all excerpts) provides a more detailed picture of population neural activity around the beat (figure 2*b*). The distribution of beat contrast (BC) values for each recorded multiunit (*N* = 77) is shown in figure 2*c*. A BC > 0 indicates that firing rates were higher on the beat than off the beat. Most multiunits show a BC > 1, and the bimodal distribution is suggestive of distinct subpopulations in the recorded data, one with BCs around 0 and the other with BCs around 0.25.

For comparison, an auditory nerve (AN) model [48] was used to estimate firing rates at the auditory nerve for 26 frequency channels covering the same range of BFs observed in our sample of cortical

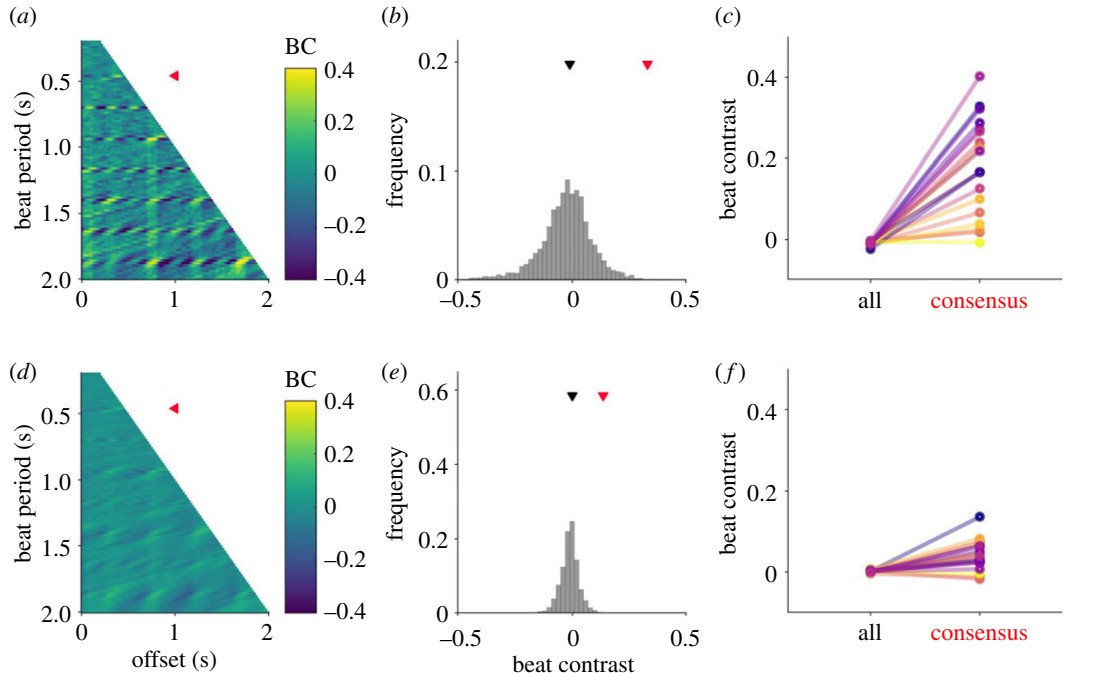

**Figure 3.** High beat contrast distinguishes the perceived beat from other possible beat interpretations. (*a*) Heatmap depicting auditory cortical beat contrasts for possible beat period (*y*-axis) and phase offset (*x*-axis) combinations between 200 ms and 2 s (beat rates of 5 Hz to 0.5 Hz) for an example musical excerpt. Red triangle indicates the consensus beat period. (*b*) Histogram of BC values in panel (*a*). The consensus BC (red triangle) is larger than the median of all BCs (black triangle). (*c*) Comparison of the each excerpt's median cortical BC (left) with its consensus BC (right). Coloured lines are excerpts 1–20, ordered from dark to light. (*d–f*) Same as panels (*a–c*), but based on AN model responses.

multiunits (figure 2*d–f*). Interestingly, the AN model would also predict higher average population firing rates on the beat than off the beat (figure 2*d*: $p < 0.001$, Wilcoxon paired signed-rank test, $N = 20$ songs). However, BCs based on the AN model, though larger than zero, are significantly smaller than cortical BCs ($p < 10^{-4}$, Wilcoxon paired signed-rank test, $N = 20$ songs).

## 3.2. A large beat contrast is a distinguishing feature of the consensus beat

While cortical firing rates in response to music are significantly stronger on the beat than off the beat, this alone does not imply that large BCs are necessarily relevant to beat perception. From a signal processing perspective, a musical excerpt could theoretically have any combination of beat period (tempo) and beat phase. If most of these possible 'musical beat interpretations' were associated with more or less equally large BCs, then a large BC would be of little value as a physiological marker for musical beat. Therefore, if a large BC is indeed relevant for the induction of musical beat, it should be selectively large at the consensus beat reported by listeners.

To test this hypothesis, BCs were computed on the population neural response (averaged across all multiunits) for the full range of possible beat period and phase combinations. For each song, possible beat periods (representing the different rates at which a listener might tap) were allowed to range from 0.2 to 2 s sampled in 20 ms steps, and phase offset was allowed to range from 0 up to the full beat period sampled in 20 ms steps. The BC was then computed for each beat interval and offset combination, resulting in 5096 total possible BC values per excerpt.

The pool of possible BCs based on the population cortical firing rate is shown in figure 3*a* for an example musical excerpt (see electronic supplementary material, figure S1 for all excerpts), and the distribution of these values is shown in figure 3*b*. Figure 3*c* compares the median BC across all beat interpretations for each musical excerpt with that excerpt's BC at the consensus beat perceived by listeners. Cortical BCs at the consensus beat were significantly larger than the median of the pool of possible BCs ($p < 10^{-4}$, Wilcoxon paired sign rank test, $N = 20$ songs). Figure 3*d–f* shows BCs based on the AN model (see electronic supplementary material, figure S2 for all excerpts). AN model BCs at the perceived beat were also significantly larger than the median of all possible BCs (figure 3*f*; $p < 0.001$,

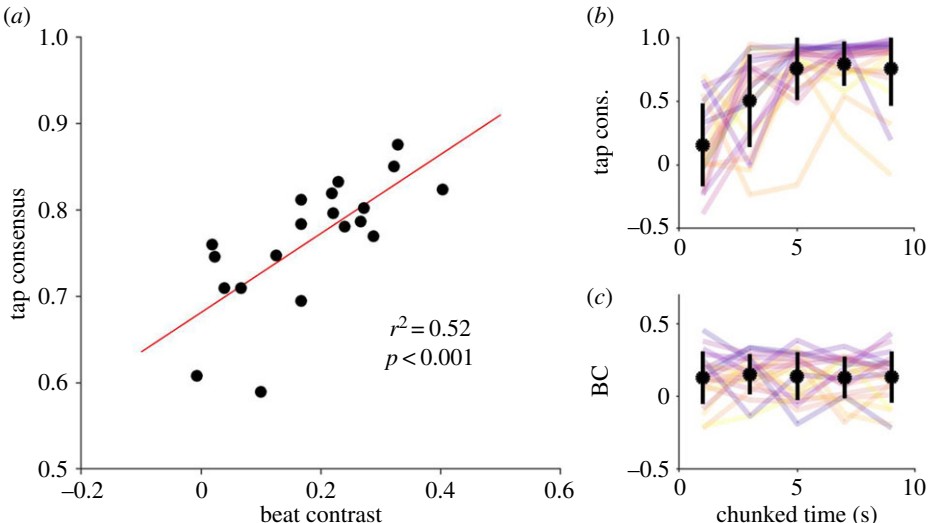

**Figure 4.** Cortical beat contrast is predictive of perceived beat ambiguity. (*a*) Auditory cortical BCs (*x*-axis) correlate strongly with tapping consensus. (*b*) Tapping consensus as it evolves over the 10 s musical excerpts. Coloured lines are excerpts 1–20, ordered from dark to light. In black are the mean across songs ± standard deviation. (*c*) Same as panel *b* but for beat contrast.

Wilcoxon paired sign rank test, $N = 20$ songs). However, there are some notable differences between cortical and AN model BCs. Cortical responses resulted in a wider range of BC values, as evidenced by the higher interquartile range of the cortical BC distributions compared to AN model BC distributions ($p < 10^{-4}$, Wilcoxon paired signed-rank test, $N = 20$ songs). Furthermore, cortical BCs at the consensus beat were above the 95th percentile of all possible BCs for 14 of the 20 musical excerpts (electronic supplementary material, figure S3), but for only 10 out of 20 excerpts for AN model responses (electronic supplementary material, figure S4). Together, these results suggest that a large BC is a feature that distinguishes the consensus beat from most other possible beat interpretations, and that two important consequences of auditory processing may be an amplification of small BCs already present at the auditory periphery, and a further restriction of the candidate beat interpretations that would result in large BCs.

## 3.3. Cortical beat contrasts are predictive of beat ambiguity

The results so far have been based on a single consensus beat interpretation for each musical excerpt. However, it is not uncommon for listeners to interpret the beat differently for a given song, or for there to be uncertainty about when the beat occurs if the beat is not very salient (see electronic supplementary material, figures S5 and S6). We have demonstrated that cortical BCs are a distinguishing feature of the consensus beat, but could they also capture how closely listeners agree on a consensus beat?

The strength of the tapping consensus was quantified for each song by calculating the correlation coefficient between the smoothed histogram of observed tap times and the 'ideal' histogram in which all 40 listeners would have tapped the consensus interpretation of beat (see *Methods*). A correlation coefficient close to 1 would therefore indicate a strong consensus across listeners. If a large BC predisposes listeners to perceive a particular interpretation of beat, then we hypothesized that songs with a larger BC at the consensus beat should show a stronger tapping consensus across listeners. Figure 4*a* confirms that, indeed, BCs in the auditory cortex correlate significantly with the strength of the tapping consensus across listeners (figure 4*a*; $p < 0.001$, Pearson correlation, $N = 20$ songs). Importantly, neither BC ($p = 0.49$) nor consensus strength ($p = 0.44$) correlated with the consensus tempo of musical excerpts (Pearson correlation, $N = 20$ songs). Figure 4*b,c* shows how BC and consensus strength develop over time. Tapping consensus strength, which is low initially, quickly reaches ceiling, indicating that listeners only needed a few seconds to find the beat. BCs on the other hand do not change systematically over time, suggesting that the correspondences described in this study between neural activity and behaviour are unlikely to be due to cortical entrainment or build-up in neural responses that are typically reported in human studies of musical beat perception [32,49–52].

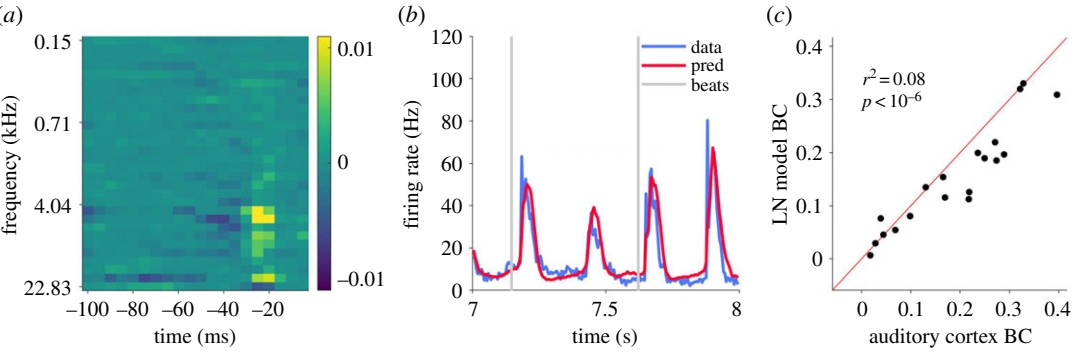

**Figure 5.** Spectrotemporal receptive field (STRF) based linear–nonlinear (LN) model captures observed cortical beat contrasts. (*a*) STRF showing a classic pattern of excitation and inhibition from an example multiunit. (*b*) Measured (blue) and LN model predicted (red) population firing rate for a 1 s segment of an example musical excerpt, consensus tap times shown by grey vertical lines. (*c*) Cortical BCs based on observed (*x*-axis) versus predicted (*y*-axis) population firing rates; each dot is one musical excerpt.

## 3.4. Spectrotemporal tuning can explain cortical beat contrast

While the physiological signals described so far exhibit many intriguing correspondences to perceived musical beat, it is nevertheless likely that the mechanisms shaping these signals reflect relatively simple temporal processing by auditory neurons rather than beat-specific processes. Indeed, we found that standard linear–nonlinear (LN) spectrotemporal receptive field (STRF) models fitted to our neural data largely reproduced the observed BCs. Each multiunit's STRF and its static sigmoid output nonlinearity were fitted using standard methodology (see *Methods*). Firing rates were then predicted using each multiunit's LN model, then averaged across multiunits to arrive at the predicted population firing rate, and from this, predicted BC values were computed for each musical excerpt.

An STRF from an example multiunit is shown in figure 5*a* (see electronic supplementary material, figure S7 for STRFs of all multiunits and their corresponding beat contrasts). This multiunit shows a preference for frequencies at and above 4 kHz, and is excited if sounds in that frequency range were heard 25 ms ago but inhibited if they occurred 40 ms ago. A multiunit's best frequency did not significantly correlate with its median beat contrast ($r = -0.21$, $p = 0.07$, Pearson correlation; see electronic supplementary material, figure S8). Predicted and observed population firing rates are in very good agreement, as shown by the example excerpt in figure 5*b*. Figure 5*c* plots observed (*x*-axis) against predicted (*y*-axis) cortical BC values for each musical excerpt. The LN model tends to underestimate BCs ($p < 0.001$, Wilcoxon paired signed-rank test, $N = 20$ songs), suggesting that there is some nonlinear process that slightly increases BC beyond processes captured by a standard LN model. However, despite this minor difference, the LN model successfully accounts for 88% of the variance in BC values for the tested musical excerpts ($p < 10^{-6}$, Pearson correlation, $N = 20$ songs), again suggesting that cortical beat contrast can be explained by spectrotemporal tuning and does not require the 'top-down' processes currently thought to drive the perception of musical beat.

## 4. Discussion

The aim of this study was to explore possible contributions of firing rate fluctuations in the auditory cortex to beat perception in real music. Our results revealed strong correspondences between the neural responses to music in the auditory cortex and the perceived location and clarity of the beat. Crucially, these effects were explained by the spectrotemporal tuning properties of recorded multiunits, indicating that they are attributable to 'bottom-up' sensory processing.

A potential weakness of this study is that comparisons were made across species. However, as the objective of this study was to explore the extent to which beat perception could be constrained by basic mechanisms of bottom-up auditory processing, this approach allowed us to test low-level contributions in a controlled situation where little to no motor or top-down activations are expected. It is therefore particularly worth noting the strong correspondences observed here between auditory cortical firing rates and tapping behaviour, and how well these correspondences could be captured by a standard LN model. Our findings suggest that fundamental, low-level mechanisms such as adaptation [13], amplitude modulation tuning [53] and temporal contrasts in STRFs play a formative role in musical

beat perception. This is consistent with the idea that the induction of beat is the result of an interaction between 'bottom-up' sensory processes and 'top-down' cognitive ones [54], perhaps through the application of learned and implicit rhythmic priors [6,7] onto an ascending sensory representation [13,53]. Our data suggest that beat perception may begin weakly at the ear, with neural activity showing stronger correspondences to behaviour as information ascends through the brainstem and primary cortical structures of the ascending auditory pathway [55,56]. Since these parts of the ascending auditory system are highly conserved across mammalian species [9–12], cross-species investigations may be a promising approach for investigating the neural signals and dynamics that underlie beat induction.

The correspondences we observe between neural responses and beat perception are probably due to the combined effect of temporal structure in the stimulus and neuronal sensitivity to long timescales. As mentioned in the introduction, Paul Fraisse already explored in the 1980s how setting accents in a series of isochronous sounds by introducing occasional differences in a sound feature (e.g. intensity, pitch) immediately evokes the perception of rhythmic groupings (see e.g. [14] for a review), and Povel & Essens [15] put forward an empirical model of beat perception that assumes that the perceived beat aligns itself maximally to such 'perceptual accents'. Our work suggests that transient increases in firing rate might be a physiological correlate of perceptual accents, since adaptation would predict larger neural responses for stimulus features that have not been experienced in the recent past. We think it is likely that the combination of excitatory and inhibitory receptive field elements in close succession, which is frequently observed in cortical neurons and well captured by LN models, heightens the sensitivity of cortical neurons to temporal contrast. Thus, our data support the notions that auditory processing gives rise to perceptual accents, for example through the detection of high temporal contrast events, and that musical beat perception arises from an interaction between perceptual accents and the temporal structure of music, which may by design be a reflection of the temporal processing capabilities of the human brain [57].

It is important to emphasize that the large responses we observed in the auditory cortex occur at points of high spectrotemporal contrast in the sound and do not selectively occur on musical beats (see figure 1e for examples of large responses that do and do not coincide with musical beats). These large fluctuations in firing rate therefore represent a preprocessing step that, we believe, constrains periodic activity at subsequent stages of processing where a single periodic interpretation of beat is selected. These subsequent stages almost certainly require feedback or recurrent connections. Computational models based on nonlinear dynamical systems, typically involving coupled oscillators, have been successful at predicting many behavioural and physiological attributes of beat perception and synchronization [25], though the biological neuronal networks that might implement these dynamics are not yet known. The cortico-basal ganglia-thalamo-cortical loop [58] may be a promising candidate since projections from auditory cortical fields to the basal ganglia have been well characterized [59], and the basal ganglia in humans have been repeatedly implicated in beat perception [24,60–62] as well as other auditory cognitive abilities [63]. We speculate that large firing rate transients in the auditory cortex in response to music could provide the rhythmic excitation required to set into motion the dynamics of this loop, and to enable entrainment of neural oscillations to the beat [32,49–52]. Caution is required, however, as there is currently some debate around what constitutes neural entrainment to auditory rhythms [64–67], and whether frequency-domain representations of rhythms and brain signals necessarily reflect beat perception [68–71].

By revealing the extent to which beat perception is shaped by bottom-up auditory processing, this study raises several questions for future investigation. One is the balance between 'bottom-up' and 'top-down' processes in beat perception. For example, some excerpts in our dataset evoked a cortical BC that was not substantially higher than the distribution of all possible BCs (e.g. excerpts 12, 13, 15; see electronic supplementary material, figure S3), and yet this consensus beat was still felt by a majority of listeners (electronic supplementary material, figure S5). This suggests that in some cases, 'top-down' processes may be more important for finding the beat, although it is worth noting that a low cortical BC typically also meant a weaker tapping consensus across listeners (figure 4a). Another aspect of musical beat perception that our cortical data do not explain is the build-up in the perception of beat, which is not accompanied by an increase in beat contrast at the sensory representation level (compare figure 4b and c). Future work could explore neural correlates for the build-up in musical beat perception, and explore which features in the sensory representation might determine the time course for this build-up. Furthermore, the perceived beat and its neural signatures can also be modulated at will by top-down attention or mental imagery of beat structure [20,50,60,72]. Further work is required to test the limits of top-down and bottom-up processes in beat perception, and to understand whether their respective influences are modulated by ambiguity in the stimulus.

Another question, given the beat-relevant precursors observed in the rodent auditory cortex, is whether rodents too might be capable of perceiving a musical beat. Growing evidence suggests that rats can be trained to discriminate isochronous from non-isochronous rhythms [73,74]. Additionally, mice are capable of licking in a way that lags an isochronous sound stimulus but continues at the correct interval even after the stimulus has stopped; in that study, primary auditory cortex was implicated as being necessary for the generation of anticipatory motor actions [75]. These studies at minimum suggest that rodents can perceive temporal structure and execute motor actions timed to an external isochronous rhythm. Future work is required to explore the limits of sensorimotor synchronization in rodents, to understand whether it extends to musical beat, and to relate these findings to behavioural and physiological data from other species. For example, it has been recently shown that non-human primates [76,77] can predictively synchronize to metronomes. Additionally, some songbird [78] and pinniped [79] species can also find and synchronize to the beat in music.

Finally, if musical beat perception is a rare and unusual ability that only humans and a few select species may possess, then it invites the question of where, functionally and anatomically, humans diverged from other species to be able to perceive musical beat. Some clues might be found in parallels that beat perception has with other abilities, particularly with the capacity for vocal learning and language [80–83]. A sensitivity to temporal regularities in the beat range may also be beneficial for detecting patterns [84,85] and segmenting auditory scenes [86,87], which relies on lower-level circuits and mechanisms such as those involved in time perception [88] and the prediction of future sensory input [89]. Cross-species investigations may be particularly suited to pinning down the key mechanisms that make beat perception, to the extent of our present understanding, characteristically human.

Ethics. All procedures were approved and licensed by the UK home office in accordance with governing legislation (ASPA 1986).

Data accessibility. Data available from the Dryad Digital Repository: https://dx.doi.org/10.5061/dryad.1k84dt2 [35]. Due to licensing restrictions, stimuli are available from: https://www.music-ir.org/mirex/wiki/2006:Audio_Beat_Tracking or from the authors on request.

Authors' contributions. V.G.R. conducted the experiments, performed the data analysis and wrote the manuscript. N.S.H. and J.W.H.S. contributed to the data analysis and critically revised the manuscript. All authors took part in designing the study, gave final approval for publication and agree to be held accountable for the work performed therein.

Competing interests. We declare we have no competing interests.

Funding. This work was supported by the Wellcome Trust (WT099750MA to V.G.R., WT076508AIA and WT108369/Z/15/Z to N.S.H.) and the Hong Kong Research Grants Council (grant nos. E-CityU101/17 and CityU11100518 to J.W.H.S.).

Acknowledgements. The authors thank Dr Israel Nelken for helpful comments on the manuscript.

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
