## [Reviewer comments · Royal Society Open Science]

Review History

RSOS-191194.R0 (Original submission)

Review form: Reviewer 1

Is the manuscript scientifically sound in its present form?

Yes

Are the interpretations and conclusions justified by the results?

Yes

Is the language acceptable?

Yes

Do you have any ethical concerns with this paper?

No

Have you any concerns about statistical analyses in this paper?

Yes

Recommendation?

Major revision is needed (please make suggestions in comments)

Comments to the Author(s)

General comments:

The authors of this study used an anesthetized rat model to measure neural responses in auditory cortex during presentations of 20 music recordings used in MIREX competitions that should evoke some sense of a musical beat. The authors find that auditory cortical neurons time-lock to the beat more than neurons simulated with an auditory nerve model, and the time-locking can be explained by a simple linear-nonlinear model based on the receptive field of the multi-units.

Overall, I think this paper is a useful addition to the rhythm perception literature, for which there are few studies on subcortical involvement in observed neural responses to musical beats. This study also makes a unique addition to this field by its use of music recordings, which has been lacking in previous studies.

I appreciate this study, but I have some concerns about the analyses, and I recommend some further analyses and discussion points.

Major comments:

My biggest concern relates to the OOR comparison between auditory cortical neurons and simulated responses from the auditory nerve. In Figure 2 it is clear that auditory cortical neurons have a lower off-beat firing rate than the auditory nerve. This could overinflate the OOR for cortical neurons relative to the auditory nerve even though the differences between on- and off-beat firing may be similar. I recommend two things: 1) Use a d-prime metric instead of a ratio, based on the mean and standard deviation of the on- and off-firing rates, and 2) Consider examining low-spontaneous auditory nerve models as well, since they may produce similar OORs (or d-prime) to the auditory cortical neurons recorded in this study.

Secondly, the range of characteristic frequencies (CF, frequencies that the neurons are tuned to) of the auditory cortical neurons is never discussed in the paper, and I recommend including that information. In particular, the range of CFs could have a major impact on the overall results. Some onset-finding algorithms had considerable success simply by looking at high frequency energy in audio recordings, because the percussive instruments often produce measurable energy in this range (take a look at the tutorial written by Bello et al, 2005, IEEE). Thus, if the auditory cortical multi-units recorded in this study have high CFs (like the one in Fig. 5), this could be contributing to the success of beat finding. I recommend examining how OOR (or d-prime) changes as a function of the CF of the multi-units – OOR might be lower for lower CF units.

Lastly, I liked the use of LN models to quantify how well spectotemporal feature extraction captured auditory cortical responses to these stimuli, but it was difficult to know what to make of these results (Figure 5). Is there any reason that the LN model shouldn't produce a similar OOR to the PSTHs? Shouldn't the LN model capture the variations in firing rate in auditory nerve activity as well, which doesn't track beats to the same extent as cortex? One way this could be clarified is by running the LN model on the simulated auditory nerve data and showing differences between the auditory cortical STRFs and the auditory nerve STRFs (or other parts of the model) in order to explain why auditory cortical neurons do so much better at beat tracking, and (hopefully) reinforce the main points of the manuscript.

Detailed comments:

line 38 – “despite beat being...” change to “despite the musical beat being...”. Also, can you provide a source for this sentence?

Line 58-59 – “...neural emphasis that predispose beat...”, perhaps: “...neural emphasis on musical beats...”

Line 61 (and throughout the introduction) – “where” is used to describe the timing of beats. Please use “when”, as in “...explain when beats...” and “...when the beat is...”

Line 61-64 – “Twenty...excerpts”. This should go in methods, not the introduction. Also, I recommend adding a table of the genre/instrumentation/tempo of each excerpt in supplemental

information, since there are only 20 excerpts, and the types of music used for stimuli could have had a major impact on the results too.

p.6 – Please move the Tapping Analysis and Consensus Beat section up to follow Stimuli in the previous page.

p.8 – Similarly, please move the Strength of Tapping Consensus section up to follow the behavioral data analysis.

Surgical Protocol – Please describe how recording sites were identified as auditory cortex. Was there some search stimulus? Did you present other stimuli besides the music excerpts?

Line 152-155 – The use of correlation coefficient here seems to depend highly on your choice of windows, both in getting the consensus tap timing and in creating the idealized tap histogram. It also wasn't clear if subject data is pooled together at some stage to get these correlation values -- is the "observed tap distribution histogram" (line 142) the pooled histogram in Figure 1c?. It would be more appropriate, I think, to quantify the variance in tap times surrounding the idealized taps. One option could be to compute the variance in tap times, averaged across idealized taps, divided by the idealized intertap interval for each track.

Line 165-167 – “noise power to signal power”, do you mean signal-to-noise ratio (SNR)? Relatedly, what is “noise power” and “signal power”? How are these computed?

Line 168 – please define what a “multi-unit” is, and relatedly, for the results, please explain what you mean by a recorded “unit”. I assume you mean a recording that consists of multiple neurons, but it would be best to be clear, especially if this is read by others who don't do extracellular, intracranial recordings.

Line 186-189 – This sentence on regularization and (I assume) 10-fold cross-validation is hard to interpret. Is this being fit to the firing rates? Is there any iteration process involved to cross-validate?

Line 223 – “150 Hz to 24 kHz”, how does this range compare to the range of CFs for the auditory cortical neurons? The results might be easier to compare if the model auditory nerve fibers were restricted to the same range.

Line 226-227 – “...very substantially and statistically significantly smaller...”, please change to “...significantly smaller...”

Line 233 & 234 – please change “beat” on both lines to “the musical beat”

Line 248-249 – The logic behind this statistical test is hard to follow. What is a “consensus OOR”? This wasn't defined earlier. Was the pool of possible OORs created for each unit and then pooled together, or was it created based on the summed firing rates across units? Lastly, using a sign test seems inappropriate here, since in Fig 2C the distribution of OORs for auditory cortical neurons is definitely bimodal. I prefer something akin to lines 255-257, where the number of OORs above the pool OORs is computed.

Line 287 – “...varied...” please change to “...correlated...”

Line 292-293 – “unlikely to be...neural responses”, please discuss further why this buildup might occur if it doesn't occur in auditory cortex

Line 300 – “...nearly 90% of the variance...” change to “...most of the variance...”

Line 318-320 – I worry this sentence could be misleading, because without the nonlinearity the linear STRF model assumes a gaussian distribution of firing rates, which is a poor assumption because the firing rates can't be less than zero. If you state the purpose of the nonlinearity in the Methods (to constrain the range of firing rates), I think you can cut this sentence.

Line 333 – “...these effects...”, you should reiterate that this does not include the relationship between tapping consensus and OOR over time (Fig. 4a and b), and discuss this later in the Discussion (see my comment on line 292-293)

Line 363-365 – “...then part...rhythmic groupings.” I don't see how this follows the previous half of the sentence, please change. In particular, I think referring to music composition here is out of context, since earlier in this paragraph you refer specifically to perceptual accents and nothing relating to how the music is constructed.

Line 413 – “...understand...” change to “...to understand...”

Review form: Reviewer 2

Is the manuscript scientifically sound in its present form?

Yes

Are the interpretations and conclusions justified by the results?

Yes

Is the language acceptable?

Yes

Do you have any ethical concerns with this paper?

No

Have you any concerns about statistical analyses in this paper?

No

Recommendation?

Major revision is needed (please make suggestions in comments)

Comments to the Author(s)

The authors describe an experiment in which firing rates in rat auditory cortex are linked to the metrical structure of rhythms, as found by human tapping preferences. More specifically, they found that 1) firing rates were higher to sounds on than off the beat, 2) the ratio between on-beat and off-beat firing was largest for the beat interpretation that humans preferred, 3) these findings were similar, but much smaller for an auditory nerve model, 4) the ratio between on-beat and off-beat firing correlated with tapping consensus amongst humans, and 5) the on-beat off-beat ratio can be explained by a spectrotemporal tuning model.

The topic of this paper is interesting. The methods used seem solid (though I am not a specialist in animal research, so I cannot comment on the methods used for the recordings in rodents). Mainly, I have some comments on the framing of the research question and results. Also, the methods lack explanation for some of the choices made (see below for specifics).

The main conceptual issue:

1. The authors describe their question as “does neural emphasis in the auditory representation of a rhythm predispose the musical beat?” (p 3, lines 49-51). Throughout the paper, the way this question is framed seems to suggest that this neural emphasis is part of the mechanisms underlying beat perception (for example: “beat perception may begin weakly at the ear”). Also, in the introduction, only theories related to beat perception are discussed. However, as the authors themselves note in the discussion, fluctuations in firing rate are “merely a preprocessing step” (line 375) and “a physiological correlate of perceptual accents” (line 360). To me, it seems that the results presented here have more to do with perceptual accenting than beat perception. The relationship between those two has been examined quite a bit before (for example by Povel and Essens in the '80). What this paper adds is a possible neural mechanism for perceptual accents (higher firing rates to accented than unaccented tones), which is an important contribution. But in my opinion, this should be framed in terms of perceptual accents, already in the introduction. That the results are based on beat locations in humans probably stems from these beat locations being based on (perceptual) accents. Therefore, in a way, the beat location is just a proxy for the location of the perceptual accents. I think the manuscript would improve if this was more clear. To solve this, I think it would be good to move the discussion about perceptual accents partly to the introduction, and explain the relationship between firing rates and perceptual accents, and perceptual accents and the beat more explicitly.

General issue:

2. The organization of the paper is a bit confusing sometimes. Some information is repeated (for example the first sentence of the results), and sometimes information in the results section refers to an analysis that was not mentioned in the methods section (for example the auditory nerve model). It would help if all methods used were present in the methods section and omitted from the results section.

Other issues:

3. Line 18: "Beat perception is regarded as a high-level ability". Is it? Later the authors describe entrainment models. Why would entrainment be "high-level"? Do you mean cortical vs. subcortical? Though beat perception has also been linked to the basal ganglia. How is this high-level?
4. Line 21: I suggest leaving out the word "surprising", given the previous paper by the same authors showing similar results for artificial stimuli.
5. Line 24: This sentence is unclear (it reads as if 40 humans reported the firing rates)
6. Line 28: I do not think your findings suggest that bottom-up processing "facilitates the extraction of a beat". Bottom-up processing is responsible for created perceptually accented tones, which in turn can induce a beat (not necessarily "facilitate" it).
7. Line 61: typo (remove "be")
8. Line 77: I do not have access to the stimuli from the MIREX competition (the website does not give me permission), so I would appreciate more information (could also be in the supplementary information). Is this instrumental music, what are the metrical structures, did any loudness normalization take place?
9. Related to this: Is there any way to link the results of the firing rates to the locations of accented vs. unaccented tones (instead of the beat as derived from the tapping data) in this dataset? For example, could you maybe provide an average waveform for the sound on the beat and the sound off the beat? Looking at Figure 1 it seems that there may have been differences in loudness or loudness-changes (onsets) for beat vs. offbeat sounds. This could very well explain your data, so it would be interesting to see whether this is the case (e.g., whether here, the accents may not just have been perceptual, but also in loudness).
10. Line 103: Could you offer some explanation as to why you chose this way of estimating the beat location from the tapping data? Why not the average time? Also, you chose the beat in such a way that beat period was constant. But if I am understanding correctly, you used real music. How do you then account for possible tempo changes?
11. Line 114: I am a bit surprised by the beat ranges. 0.7 and 3.7 Hz seem well outside of the rate at which humans normally can perceive a beat. How do you explain this? Could it be that the tapping was at subdivisions or higher-level meter instead of the beat? How do you account for this in the data?
12. Line 141: Why did you use this metric for tapping consensus? Why not just the spread in tapping times?
13. Line 157: Was data averaged over the three rats? Probably, but maybe mention this somewhere.
14. Line 222: The rationale for the auditory nerve model only becomes clear later in the paper. This could be added to the methods, and maybe even the introduction, to explain that you will compare your results to this model, and why.
15. Line 249: typo (Figure 3C is now 3D?)
16. Line 365-368: This sentence was unclear to me, and it does not seem to relate to the previous.
17. Line 370: "Interaction between perceptual accents and temporal structure" is a bit unclear. Do you maybe mean the "temporal structure of the accents"?
18. Line 416: Beat perception is not an ability that only humans possess. Several species have been shown to be capable of beat perception (Snowball, see Patel et al., 2009; Ronan, see Cook et al., 2013), and recently, Hugo Merchant's lab also showed that rhesus monkeys, with the right incentive, can be taught to tap to a metronome, see Gamez et al., 2018. While you could question whether the latter is really beat perception, it would be good to nuance the statement in this sentence a bit.

Decision letter (RSOS-191194.R0)

17-Oct-2019

Dear Ms Rajendran,

The editors assigned to your paper ("Auditory Cortical Representation of Music Favours the Perceived Beat") have now received comments from reviewers. We would like you to revise your paper in accordance with the referee and Associate Editor suggestions which can be found below (not including confidential reports to the Editor). Please note this decision does not guarantee eventual acceptance.

Please submit a copy of your revised paper before 09-Nov-2019. Please note that the revision deadline will expire at 00.00am on this date. If we do not hear from you within this time then it will be assumed that the paper has been withdrawn. In exceptional circumstances, extensions may be possible if agreed with the Editorial Office in advance. We do not allow multiple rounds of revision so we urge you to make every effort to fully address all of the comments at this stage. If deemed necessary by the Editors, your manuscript will be sent back to one or more of the original reviewers for assessment. If the original reviewers are not available, we may invite new reviewers.

- Data accessibility

<http://datadryad.org/submit?journalID=RSOS&manu=RSOS-191194>

- **Competing interests**

- **Authors' contributions**

- **Acknowledgements**

- **Funding statement**

Kind regards,
Anita Kristiansen
Editorial Coordinator
Royal Society Open Science
openscience@royalsociety.org

on behalf of Dr Carolyn McGettigan (Associate Editor) and Essi Viding (Subject Editor)
openscience@royalsociety.org

Associate Editor's comments (Dr Carolyn McGettigan):

Associate Editor:

Comments to the Author:

I have received reviews from 2 experts in the field, who both find your paper interesting but have raised some concerns regarding aspects of the analyses and the conceptual framing of your research question. I recommend that you address these concerns in a revision of your article, to include a point-by-point response to the reviewers.

Reviewers' Comments to Author:

Reviewer: 1

Comments to the Author(s)

General comments:

The authors of this study used an anesthetized rat model to measure neural responses in auditory cortex during presentations of 20 music recordings used in MIREX competitions that should evoke some sense of a musical beat. The authors find that auditory cortical neurons time-lock to the beat more than neurons simulated with an auditory nerve model, and the time-locking can be explained by a simple linear-nonlinear model based on the receptive field of the multi-units.

Overall, I think this paper is a useful addition to the rhythm perception literature, for which there are few studies on subcortical involvement in observed neural responses to musical beats. This study also makes a unique addition to this field by its use of music recordings, which has been lacking in previous studies.

I appreciate this study, but I have some concerns about the analyses, and I recommend some further analyses and discussion points.

Major comments:

My biggest concern relates to the OOR comparison between auditory cortical neurons and simulated responses from the auditory nerve. In Figure 2 it is clear that auditory cortical neurons have a lower off-beat firing rate than the auditory nerve. This could overinflate the OOR for cortical neurons relative to the auditory nerve even though the differences between on- and off-beat firing may be similar. I recommend two things: 1) Use a d-prime metric instead of a ratio, based on the mean and standard deviation of the on- and off-firing rates, and 2) Consider examining low-spontaneous auditory nerve models as well, since they may produce similar OORs (or d-prime) to the auditory cortical neurons recorded in this study.

Secondly, the range of characteristic frequencies (CF, frequencies that the neurons are tuned to) of the auditory cortical neurons is never discussed in the paper, and I recommend including that information. In particular, the range of CFs could have a major impact on the overall results. Some onset-finding algorithms had considerable success simply by looking at high frequency energy in audio recordings, because the percussive instruments often produce measurable energy in this range (take a look at the tutorial written by Bello et al, 2005, IEEE). Thus, if the auditory cortical multi-units recorded in this study have high CFs (like the one in Fig. 5), this could be contributing to the success of beat finding. I recommend examining how OOR (or d-prime) changes as a function of the CF of the multi-units – OOR might be lower for lower CF units.

Lastly, I liked the use of LN models to quantify how well spectotemporal feature extraction captured auditory cortical responses to these stimuli, but it was difficult to know what to make of these results (Figure 5). Is there any reason that the LN model shouldn't produce a similar OOR to the PSTHs? Shouldn't the LN model capture the variations in firing rate in auditory nerve activity as well, which doesn't track beats to the same extent as cortex? One way this could be clarified is by running the LN model on the simulated auditory nerve data and showing differences between the auditory cortical STRFs and the auditory nerve STRFs (or other parts of the model) in order to explain why auditory cortical neurons do so much better at beat tracking, and (hopefully) reinforce the main points of the manuscript.

Detailed comments:

line 38 – “despite beat being...” change to “despite the musical beat being...”. Also, can you provide a source for this sentence?

Line 58-59 – “...neural emphasis that predispose beat...”, perhaps: “...neural emphasis on musical beats...”

Line 61 (and throughout the introduction) – “where” is used to describe the timing of beats. Please use “when”, as in “...explain when beats...” and “...when the beat is...”

Line 61-64 – “Twenty...excerpts”. This should go in methods, not the introduction. Also, I recommend adding a table of the genre/instrumentation/tempo of each excerpt in supplemental information, since there are only 20 excerpts, and the types of music used for stimuli could have had a major impact on the results too.

p.6 – Please move the Tapping Analysis and Consensus Beat section up to follow Stimuli in the previous page.

p.8 – Similarly, please move the Strength of Tapping Consensus section up to follow the behavioral data analysis.

Surgical Protocol – Please describe how recording sites were identified as auditory cortex. Was there some search stimulus? Did you present other stimuli besides the music excerpts?

Line 152-155 – The use of correlation coefficient here seems to depend highly on your choice of windows, both in getting the consensus tap timing and in creating the idealized tap histogram. It also wasn't clear if subject data is pooled together at some stage to get these correlation values -- is the "observed tap distribution histogram" (line 142) the pooled histogram in Figure 1c?. It would be more appropriate, I think, to quantify the variance in tap times surrounding the idealized taps. One option could be to compute the variance in tap times, averaged across idealized taps, divided by the idealized intertap interval for each track.

Line 165-167 – “noise power to signal power”, do you mean signal-to-noise ratio (SNR)? Relatedly, what is “noise power” and “signal power”? How are these computed?

Line 168 – please define what a “multi-unit” is, and relatedly, for the results, please explain what you mean by a recorded “unit”. I assume you mean a recording that consists of multiple neurons, but it would be best to be clear, especially if this is read by others who don't do extracellular, intracranial recordings.

Line 186-189 – This sentence on regularization and (I assume) 10-fold cross-validation is hard to interpret. Is this being fit to the firing rates? Is there any iteration process involved to cross-validate?

Line 223 – “150 Hz to 24 kHz”, how does this range compare to the range of CFs for the auditory cortical neurons? The results might be easier to compare if the model auditory nerve fibers were restricted to the same range.

Line 226-227 – “...very substantially and statistically significantly smaller...”, please change to “...significantly smaller...”

Line 233 & 234 – please change “beat” on both lines to “the musical beat”

Line 248-249 – The logic behind this statistical test is hard to follow. What is a “consensus OOR”? This wasn't defined earlier. Was the pool of possible OORs created for each unit and then pooled together, or was it created based on the summed firing rates across units? Lastly, using a sign test seems inappropriate here, since in Fig 2C the distribution of OORs for auditory cortical neurons is definitely bimodal. I prefer something akin to lines 255-257, where the number of OORs above the pool OORs is computed.

Line 287 – “...varied...” please change to “...correlated...”

Line 292-293 – “unlikely to be...neural responses”, please discuss further why this buildup might occur if it doesn't occur in auditory cortex

Line 300 – “...nearly 90% of the variance...” change to “...most of the variance...”

Line 318-320 – I worry this sentence could be misleading, because without the nonlinearity the linear STRF model assumes a gaussian distribution of firing rates, which is a poor assumption because the firing rates can't be less than zero. If you state the purpose of the nonlinearity in the Methods (to constrain the range of firing rates), I think you can cut this sentence.

Line 333 – “...these effects...”, you should reiterate that this does not include the relationship between tapping consensus and OOR over time (Fig. 4a and b), and discuss this later in the Discussion (see my comment on line 292-293)

Line 363-365 – “...then part...rhythmic groupings.” I don't see how this follows the previous half of the sentence, please change. In particular, I think referring to music composition here is out of context, since earlier in this paragraph you refer specifically to perceptual accents and nothing relating to how the music is constructed.

Line 413 – “...understand...” change to “...to understand...”

Reviewer: 2

Comments to the Author(s)

The authors describe an experiment in which firing rates in rat auditory cortex are linked to the metrical structure of rhythms, as found by human tapping preferences. More specifically, they

found that 1) firing rates were higher to sounds on than off the beat, 2) the ratio between on-beat and off-beat firing was largest for the beat interpretation that humans preferred, 3) these findings were similar, but much smaller for an auditory nerve model, 4) the ratio between on-beat and off-beat firing correlated with tapping consensus amongst humans, and 5) the on-beat off-beat ratio can be explained by a spectrotemporal tuning model.

The topic of this paper is interesting. The methods used seem solid (though I am not a specialist in animal research, so I cannot comment on the methods used for the recordings in rodents). Mainly, I have some comments on the framing of the research question and results. Also, the methods lack explanation for some of the choices made (see below for specifics).

The main conceptual issue:

1. The authors describe their question as “does neural emphasis in the auditory representation of a rhythm predispose the musical beat?” (p 3, lines 49-51). Throughout the paper, the way this question is framed seems to suggest that this neural emphasis is part of the mechanisms underlying beat perception (for example: “beat perception may begin weakly at the ear”). Also, in the introduction, only theories related to beat perception are discussed. However, as the authors themselves note in the discussion, fluctuations in firing rate are “merely a preprocessing step” (line 375) and “a physiological correlate of perceptual accents” (line 360). To me, it seems that the results presented here have more to do with perceptual accenting than beat perception. The relationship between those two has been examined quite a bit before (for example by Povel and Essens in the '80). What this paper adds is a possible neural mechanism for perceptual accents (higher firing rates to accented than unaccented tones), which is an important contribution. But in my opinion, this should be framed in terms of perceptual accents, already in the introduction. That the results are based on beat locations in humans probably stems from these beat locations being based on (perceptual) accents. Therefore, in a way, the beat location is just a proxy for the location of the perceptual accents. I think the manuscript would improve if this was more clear. To solve this, I think it would be good to move the discussion about perceptual accents partly to the introduction, and explain the relationship between firing rates and perceptual accents, and perceptual accents and the beat more explicitly.

General issue:

2. The organization of the paper is a bit confusing sometimes. Some information is repeated (for example the first sentence of the results), and sometimes information in the results section refers to an analysis that was not mentioned in the methods section (for example the auditory nerve model). It would help if all methods used were present in the methods section and omitted from the results section.

Other issues:

3. Line 18: “Beat perception is regarded as a high-level ability”. Is it? Later the authors describe entrainment models. Why would entrainment be “high-level”? Do you mean cortical vs. subcortical? Though beat perception has also been linked to the basal ganglia. How is this high-level?

4. Line 21: I suggest leaving out the word “surprising”, given the previous paper by the same authors showing similar results for artificial stimuli.

5. Line 24: This sentence is unclear (it reads as if 40 humans reported the firing rates)

6. Line 28: I do not think your findings suggest that bottom-up processing “facilitates the extraction of a beat”. Bottom-up processing is responsible for created perceptually accented tones, which in turn can induce a beat (not necessarily “facilitate” it).

7. Line 61: typo (remove “be”)

8. Line 77: I do not have access to the stimuli from the MIREX competition (the website does not give me permission), so I would appreciate more information (could also be in the supplementary information). Is this instrumental music, what are the metrical structures, did any loudness normalization take place?

9. Related to this: Is there any way to link the results of the firing rates to the locations of accented vs. unaccented tones (instead of the beat as derived from the tapping data) in this dataset? For

example, could you maybe provide an average waveform for the sound on the beat and the sound off the beat? Looking at Figure 1 it seems that there may have been differences in loudness or loudness-changes (onsets) for beat vs. offbeat sounds. This could very well explain your data, so it would be interesting to see whether this is the case (e.g., whether here, the accents may not just have been perceptual, but also in loudness).

10. Line 103: Could you offer some explanation as to why you chose this way of estimating the beat location from the tapping data? Why not the average time? Also, you chose the beat in such a way that beat period was constant. But if I am understanding correctly, you used real music. How do you then account for possible tempo changes?

11. Line 114: I am a bit surprised by the beat ranges. 0.7 and 3.7 Hz seem well outside of the rate at which humans normally can perceive a beat. How do you explain this? Could it be that the tapping was at subdivisions or higher-level meter instead of the beat? How do you account for this in the data?

12. Line 141: Why did you use this metric for tapping consensus? Why not just the spread in tapping times?

13. Line 157: Was data averaged over the three rats? Probably, but maybe mention this somewhere.

14. Line 222: The rationale for the auditory nerve model only becomes clear later in the paper. This could be added to the methods, and maybe even the introduction, to explain that you will compare your results to this model, and why.

15. Line 249: typo (Figure 3C is now 3D?)

16. Line 365-368: This sentence was unclear to me, and it does not seem to relate to the previous.

17. Line 370: "Interaction between perceptual accents and temporal structure" is a bit unclear. Do you maybe mean the "temporal structure of the accents"?

18. Line 416: Beat perception is not an ability that only humans possess. Several species have been shown to be capable of beat perception (Snowball, see Patel et al., 2009; Ronan, see Cook et al., 2013), and recently, Hugo Merchant's lab also showed that rhesus monkeys, with the right incentive, can be taught to tap to a metronome, see Gamez et al., 2018. While you could question whether the latter is really beat perception, it would be good to nuance the statement in this sentence a bit.

Author's Response to Decision Letter for (RSOS-191194.R0)

See Appendix A.

RSOS-191194.R1 (Revision)

Review form: Reviewer 1

Is the manuscript scientifically sound in its present form?

Yes

Are the interpretations and conclusions justified by the results?

Yes

Is the language acceptable?

Yes

Do you have any ethical concerns with this paper?

No

Have you any concerns about statistical analyses in this paper?

No

Recommendation?

Accept with minor revision (please list in comments)

Comments to the Author(s)

The edits the authors have made have considerably improved the manuscript. However, I spotted several more minor issues that I think should be corrected:

Line 27 -- "interpretations of beat" change to "interpretations of the beat"

Line 41 -- "despite musical beat being", maybe "despite musical beats being" or "despite the musical beat being"

Methods -- Generally the flow of the methods section is much better than in the previous version. However, the "On-beat, and off-beat, and beat contrast (BC)" section is out of place because it refers to analyses of neural activity before the method of collecting and preprocessing the neural activity is described. This section should go after preprocessing. Also, I think the title of the section could be changed to "neural data analysis", or at the very least please remove the first "and" in the title.

Line 167 -- "...but are e.g. twice the speed..." Please be more direct rather than using "e.g." "...a multiple of the speed..." or "...two or three times the speed..." could work, whichever is more accurate.

Line 232 -- "in figures...", are you referring to Fig S7?

Line 340-341 -- "...are unlikely to be due...human studies of musical beat perception." Please provide a citation.

Line 360-362 -- Thank you for including the STRFs for all of the units in Fig. S7 and their beat contrasts, it is very informative. However, I think you should also include the plot you provided in the reviewer comments of beat contrast vs CF as another supplemental figure, because the relationship (or specifically the lack of a relationship) between beat contrast and CF can be hard to decipher from S7 alone, and it also shows the distribution of CFs of the population. When you include that figure, I recommend making the dots bigger so they are easier to see.

Fig. 5a -- please include a colorbar, something like what you did for Fig. S7 is sufficient.

Line 422-423 -- "...do not only occur periodically at the beat (see Fig. 1E)." This is a bit hard to decipher, do you mean that increases in activity can occur off the beat as well?

Line 458 -- "...perceiving musical beat" change to "perceiving a musical beat".

Review form: Reviewer 2**Is the manuscript scientifically sound in its present form?**

Yes

Are the interpretations and conclusions justified by the results?

Yes

Is the language acceptable?

Yes

Do you have any ethical concerns with this paper?

No

Have you any concerns about statistical analyses in this paper?

No

Recommendation?

Accept as is

Comments to the Author(s)

The authors have addressed all my comments, and the paper has been improved substantially.

Decision letter (RSOS-191194.R1)

28-Jan-2020

Dear Ms Rajendran:

On behalf of the Editors, I am pleased to inform you that your Manuscript RSOS-191194.R1 entitled "Auditory Cortical Representation of Music Favours the Perceived Beat" has been accepted for publication in Royal Society Open Science subject to minor revision in accordance with the referee suggestions. Please find the referees' comments at the end of this email.

The reviewers and Subject Editor have recommended publication, but also suggest some minor revisions to your manuscript. Therefore, I invite you to respond to the comments and revise your manuscript.

- Ethics statement

- Data accessibility

<http://datadryad.org/submit?journalID=RSOS&manu=RSOS-191194.R1>

- **Competing interests**

- **Authors' contributions**

- **Acknowledgements**

- **Funding statement**

Because the schedule for publication is very tight, it is a condition of publication that you submit the revised version of your manuscript before 06-Feb-2020. Please note that the revision deadline will expire at 00.00am on this date. If you do not think you will be able to meet this date please let me know immediately.

1) A text file of the manuscript (tex, txt, rtf, docx or doc), references, tables (including captions) and figure captions. Do not upload a PDF as your "Main Document".

- 2) A separate electronic file of each figure (EPS or print-quality PDF preferred (either format should be produced directly from original creation package), or original software format)
- 3) Included a 100 word media summary of your paper when requested at submission. Please ensure you have entered correct contact details (email, institution and telephone) in your user account
- 4) Included the raw data to support the claims made in your paper. You can either include your data as electronic supplementary material or upload to a repository and include the relevant doi within your manuscript
- 5) All supplementary materials accompanying an accepted article will be treated as in their final form. Note that the Royal Society will neither edit nor typeset supplementary material and it will be hosted as provided. Please ensure that the supplementary material includes the paper details where possible (authors, article title, journal name).

on behalf of Prof Essi Viding (Subject Editor)
openscience@royalsociety.org

Associate Editor Comments to Author:

Thank you for taking the time to revise the manuscript. As you'll see, both reviewers are broadly positively inclined towards the work, though one offers a number of recommendations that would likely improve the manuscript still further. Please can you provide a response to these recommendations - both in a point-by-point response document and in a finally revised manuscript document?

Reviewer comments to Author:

Reviewer: 1

Comments to the Author(s)

The edits the authors have made have considerably improved the manuscript. However, I spotted several more minor issues that I think should be corrected:

Line 27 -- "interpretations of beat" change to "interpretations of the beat"

Line 41 -- "despite musical beat being", maybe "despite musical beats being" or "despite the musical beat being"

Methods -- Generally the flow of the methods section is much better than in the previous version. However, the "On-beat, and off-beat, and beat contrast (BC)" section is out of place because it

refers to analyses of neural activity before the method of collecting and preprocessing the neural activity is described. This section should go after preprocessing. Also, I think the title of the section could be changed to "neural data analysis", or at the very least please remove the first "and" in the title.

Line 167 -- "...but are e.g. twice the speed..." Please be more direct rather than using "e.g." "...a multiple of the speed..." or "...two or three times the speed..." could work, whichever is more accurate.

Line 232 -- "in figures...", are you referring to Fig S7?

Line 340-341 -- "...are unlikely to be due...human studies of musical beat perception." Please provide a citation.

Line 360-362 -- Thank you for including the STRFs for all of the units in Fig. S7 and their beat contrasts, it is very informative. However, I think you should also include the plot you provided in the reviewer comments of beat contrast vs CF as another supplemental figure, because the relationship (or specifically the lack of a relationship) between beat contrast and CF can be hard to decipher from S7 alone, and it also shows the distribution of CFs of the population. When you include that figure, I recommend making the dots bigger so they are easier to see.

Fig. 5a -- please include a colorbar, something like what you did for Fig. S7 is sufficient.

Line 422-423 -- "...do not only occur periodically at the beat (see Fig. 1E)." This is a bit hard to decipher, do you mean that increases in activity can occur off the beat as well?

Line 458 -- "...perceiving musical beat" change to "perceiving a musical beat".

Reviewer: 2

Comments to the Author(s)

The authors have addressed all my comments, and the paper has been improved substantially.

Author's Response to Decision Letter for (RSOS-191194.R1)

See Appendix B.

Decision letter (RSOS-191194.R2)

03-Feb-2020

Dear Ms Rajendran,

It is a pleasure to accept your manuscript entitled "Auditory Cortical Representation of Music Favours the Perceived Beat" in its current form for publication in Royal Society Open Science. The comments of the reviewer(s) who reviewed your manuscript are included at the foot of this letter.

on behalf of Prof Essi Viding (Subject Editor)
openscience@royalsociety.org

Appendix A

5 December 2019

To the reviewers:

We would like to thank you both for taking the time and effort to thoughtfully review our work. We feel that addressing your thorough and constructive critiques has allowed us to substantially improve the quality of this work, and we are grateful for the input. Below we address each comment one by one.

Sincerely,

Vani, Nicol, Jan

Reviewer: 1

Comments to the Author(s)

General comments:

The authors of this study used an anesthetized rat model to measure neural responses in auditory cortex during presentations of 20 music recordings used in MIREX competitions that should evoke some sense of a musical beat. The authors find that auditory cortical neurons time-lock to the beat more than neurons simulated with an auditory nerve model, and the time-locking can be explained by a simple linear-nonlinear model based on the receptive field of the multi-units.

Overall, I think this paper is a useful addition to the rhythm perception literature, for which there are few studies on subcortical involvement in observed neural responses to musical beats. This study also makes a unique addition to this field by its use of music recordings, which has been lacking in previous studies.

I appreciate this study, but I have some concerns about the analyses, and I recommend some further analyses and discussion points.

Major comments:

My biggest concern relates to the OOR comparison between auditory cortical neurons and simulated responses from the auditory nerve. In Figure 2 it is clear that auditory cortical neurons have a lower off-beat firing rate than the auditory nerve. This could overinflate the OOR for cortical neurons relative to the auditory nerve even though the

differences between on- and off-beat firing may be similar. I recommend two things: 1) Use a d-prime metric instead of a ratio, based on the mean and standard deviation of the on- and off-firing rates, and 2) Consider examining low-spontaneous auditory nerve models as well, since they may produce similar OORs (or d-prime) to the auditory cortical neurons recorded in this study.

These are good points raised by the reviewer, and we have largely incorporated both suggestions into the revised manuscript, with some modifications.

To the reviewer's first suggestion, we agree with the possibility that division by a small number could overinflate the OOR. However, a d-prime metric measures something rather different from what we are trying to document here. By design, d' measures detectability. In our study, all musical excerpts were presented at sound levels well above threshold, and both on- and off-beat segments of music should be very easily detectable, and their difference in detectability is bound to be very small. The hypothesis we are trying to test here is not about detectability, it is about "salience" or "contrast," and the metric used to quantify neural responses needs to reflect this. In order to take on board the reviewer's concern that our original simple on/off ratio could become unduly inflated by small values in the denominator, we completely re-analysed our data using a Michelson contrast metric of the type that is almost universally used to quantify contrast in visual stimuli. In vision science, Michelson contrast is defined as $(I_{\max} - I_{\min}) / (I_{\max} + I_{\min})$, where I_{\max} and I_{\min} respectively are the light intensity of the brightest and darkest parts of the image. In our revision we analogously define "beat contrast" as $(ON - OFF) / (ON + OFF)$, where ON and OFF are the neural responses "on" or "off" the beat respectively. Since the denominator is the sum of both ON and OFF, a spuriously small OFF value cannot shrink it, thus guarding against the reviewer's apt concern that dividing by almost zero could grossly inflate the computed values. Indeed Michelson contrast values are bounded between -1 and 1, where 0 implies $ON = OFF$, and 1 implies $ON \gg OFF$. We believe our Michelson contrast metric is more robust than the one we proposed originally and are grateful to the reviewer for prompting us to revisit this.

We have also adopted the reviewer's second suggestion to use a low-spontaneous rate auditory nerve model, and the results of this model are now what are reported in the revised paper. While we were doing this, we also took the opportunity to make the CFs of the AN model and STRF model consistent with each other, which was not the case in the original manuscript. Neither of these adjustments change the bottom line of the results.

Secondly, the range of characteristic frequencies (CF, frequencies that the neurons are tuned to) of the auditory cortical neurons is never discussed in the paper, and I recommend including that information. In particular, the range of CFs could have a major impact on the overall results. Some onset-finding algorithms had considerable success simply by looking at high frequency energy in audio recordings, because the percussive instruments often produce measurable energy in this range (take a look at the tutorial written by Bello et al, 2005, IEEE). Thus, if the auditory cortical multi-units recorded in this study have high CFs (like the one in Fig. 5), this could be contributing to the success of beat finding. I recommend examining how OOR (or d-prime) changes as a function of the CF of the multi-units – OOR might be lower for lower CF units.

Again, a very good suggestion. Below is a plot of CF against beat contrast showing a poor correlation between the two ($r = -0.21$, $p=0.07$, Pearson correlation), and if there is a weak trend, it is in the opposite direction of what would be predicted if beat contrast was just a trivial reflection of onset detection at high frequencies.

Our intuition is that beat contrast actually has more to do with temporal contrast sensitivity rather than BF itself. We have added an additional supplementary figure (Fig. S7) with the STRFs for all recorded multiunits, presented in order of increasing beat contrast. We believe this figure allows the reader to observe and consider the features in the STRF that may be driving a high beat contrast, which to our eye appears to be a strong excitation and a prolonged period of inhibition at whichever frequency the multiunit is most sensitive to. We believe this adds additional insight to the paper and we thank the reviewer for suggesting this analysis. We

now mention in the results section the lack of correlation between a multiunit's BF and beat contrast and refer to this supplementary figure:

The best frequency of multiunits did not significantly correlate with its beat contrast ($r=-0.21$, $p=0.07$, Pearson correlation; see Supplementary Fig. S7 for STRFs of all multiunits and their corresponding beat contrasts).

Lastly, I liked the use of LN models to quantify how well spectotemporal feature extraction captured auditory cortical responses to these stimuli, but it was difficult to know what to make of these results (Figure 5). Is there any reason that the LN model shouldn't produce a similar OOR to the PSTHs? Shouldn't the LN model capture the variations in firing rate in auditory nerve activity as well, which doesn't track beats to the same extent as cortex? One way this could be clarified is by running the LN model on the simulated auditory nerve data and showing differences between the auditory cortical STRFs and the auditory nerve STRFs (or other parts of the model) in order to explain why auditory cortical neurons do so much better at beat tracking, and (hopefully) reinforce the main points of the manuscript.

We agree with the reviewer that the interpretation of the LN model results needs some additional clarification. It's true that the LN model should capture the AN model results too (but the STRFs that result would look different from those in the auditory cortex... we expect that the frequency tuning would be much narrower and the excitation latencies much shorter). The point of this analysis was really only to demonstrate that the beat contrasts we observe in the auditory cortex and their correspondences to perception can be adequately explained by basic spectrotemporal tuning properties of cells and do not require "top-down" influences or entrainment, which are processes currently thought to drive the perception of musical beat. We have added to the last sentence of the results section to clarify this:

...the LN model successfully accounts for 88% of the variance in BC values for the tested musical excerpts ($p<10^{-6}$, Pearson correlation, $N=20$ songs), suggesting that cortical beat contrast can be explained by spectrotemporal tuning and does not require the "top-down" processes currently thought to drive the perception of musical beat.

However, this and the reviewer's previous question made us wonder whether which features in the STRF – spectral, temporal, or the combination of the two – are really the ones that drive

beat contrast. We believe our additional supplementary figure showing STRFs and corresponding beat contrast values should provide some insight into this question.

Detailed comments:

line 38 – “despite beat being...” change to “despite the musical beat being...”. Also, can you provide a source for this sentence?

Both done.

Line 58-59 – “...neural emphasis that predispose beat...”, perhaps: “...neural emphasis on musical beats...”

We have changed this to read:

“However, the hypothesis that auditory processing creates points of neural emphasis that shape the perception of musical beat must first pass a crucial test...”

– this is just our hypothesis, not a strong claim. But it is also important to express that, the way we see it, this neural emphasis drives where we hear the beat and not the other way around (as suggested by the reviewer), where beats are veridical and they simply happen to have a neural emphasis on them.

Line 61 (and throughout the introduction) – “where” is used to describe the timing of beats. Please use “when”, as in “...explain when beats...” and “...when the beat is...”

Done.

Line 61-64 – “Twenty...excerpts”. This should go in methods, not the introduction. Also, I recommend adding a table of the genre/instrumentation/tempo of each excerpt in supplemental information, since there are only 20 excerpts, and the types of music used for stimuli could have had a major impact on the results too.

Done. We have also added the suggested table to the supplementary info and refer to it in the Methods.

p.6 – Please move the Tapping Analysis and Consensus Beat section up to follow Stimuli

in the previous page.

p.8 – Similarly, please move the Strength of Tapping Consensus section up to follow the behavioral data analysis.

Both done, flows better now. Thanks for this suggestion.

Surgical Protocol – Please describe how recording sites were identified as auditory cortex. Was there some search stimulus? Did you present other stimuli besides the music excerpts?

We have added the following sentences to clarify how we identified the recording site as auditory cortex:

The probe was first positioned over the auditory cortex based on anatomical coordinates and then inserted into the brain in a medio-lateral orientation if possible until all channels were inside the brain. After a few minutes, a search stimulus consisting of broadband noise bursts was played to check that recording sites were driven by sound. Next, frequency response areas (FRAs) were measured to check that channels were frequency tuned, and then the music stimuli were presented.

Line 152-155 – The use of correlation coefficient here seems to depend highly on your choice of windows, both in getting the consensus tap timing and in creating the idealized tap histogram. It also wasn't clear if subject data is pooled together at some stage to get these correlation values -- is the "observed tap distribution histogram" (line 142) the pooled histogram in Figure 1c?. It would be more appropriate, I think, to quantify the variance in tap times surrounding the idealized taps. One option could be to compute the variance in tap times, averaged across idealized tap.s, divided by the idealized intertap interval for each track.

Subject data were indeed pooled to generate the histograms for our tapping consensus measure – we now state this explicitly. Figure 1c shows a smoothed version of the histogram. We tested a range of smoothing windows to make sure that the metric was not very sensitive to window width.

The reviewer's suggestion to simply compute the variance in tap times around idealised or consensus beat times is one that we had thought hard about but discarded for the following reason: it is quite common for some observers to tap the beat at twice the speed of others, but how "wrong" are these listeners? Supplementary Figure S5, Excerpt 10 is a

particularly clear example of this. It does not seem appropriate to deem listeners tapping at 2x the consensus tempo to be exhibiting a “high degree of uncertainty” about where the beat should be since arguably, it is just an alternative (precise!) interpretation of the beat, where half the taps are “spot on” and the other half would have maximal “error” with respect to the consensus beat. This would therefore weigh extremely heavily in a variance measure, which would add the greatest possible penalty, half the beat interval squared, for every extra beat tapped by a listener tapping at 2x the consensus rate. Consequently, a very large value for a simple variance measure such as that proposed by the reviewer could easily be misinterpreted to mean that the listeners were uncertain about where the beat should be when in reality all listeners tap every beat with very high precision, but some at twice the speed of others. We chose our correlation based consensus beat measure because it avoids grossly over inflating any error terms introduced by minority beat interpretations which are at 2x (or 3x, in some cases) the speed of the majority of listeners. We now mention this in the revised Methods section:

We chose this particular correlation measure (instead of e.g. simply quantifying the variance of observed tap times around consensus beats) in order to avoid excessively penalizing minority but meter-related beat interpretations that are reported with high precision, but at e.g. twice the speed of the majority of listeners (see Supplementary Figure S5, Excerpt 10, for example).

Line 165-167 – “noise power to signal power”, do you mean signal-to-noise ratio (SNR)? Relatedly, what is “noise power” and “signal power”? How are these computed?

Our original manuscript was admittedly too terse in describing these metrics, and we have improved this in the revision. Noise power and signal power are ANOVA-style explained variance vs residual variance measures introduced by Sahani and Linden (2003), which are computed in order to quantify how well the mean PSTH predicts single trial PSTHs for repeated presentations of a given stimulus. If a neuron responds very reliably, single trial PSTHs will all be similar to each other and to the mean PSTH, and the residual “noise power” will be small compared to the “signal power” (taken to be the variance of the mean PSTH). This “Sahani measure” of response reliability is quite commonly used (their 2003 paper has 125 citations). We picked the cut-off criterion value of 40 to conform with previous practice (e.g. Rabinowitz et al J Neurosci 2012). This is now clarified in the revised manuscript.

Line 168 – please define what a "multi-unit" is, and relatedly, for the results, please explain what you mean by recorded "unit". I assume you mean a recording that consists of multiple neurons, but it would be best to be clear, especially if this is read by others who don't do extracellular, intracranial recordings.

Good suggestion. We have added the following sentence to the methods:

Each resulting cluster of spikes, which putatively originates from a small population of neurons near a recording site, is termed a multiunit.

And for consistency, we have also changed "unit" to "multiunit" throughout the paper.

Line 186-189 – This sentence on regularization and (I assume) 10-fold cross-validation is hard to interpret. Is this being fit to the firing rates? Is there any iteration process involved to cross-validate?

The LN-STRF fitting procedure is rather involved, and the details given in our original manuscript were too sparse and confusing. However, we followed precisely the method described in Ref 46 (Willmore, et al. 2016 J Neurosci), which is described in considerable detail there. Instead of duplicating the description from that reference, we now refer the reader to that open source paper.

Line 223 – "150 Hz to 24 kHz", how does this range compare to the range of CFs for the auditory cortical neurons? The results might be easier to compare if the model auditory nerve fibers were restricted to the same range.

Fair question. As it happens, the BFs of the auditory cortical neurons covered a similarly wide range, from 300 Hz to 22833 Hz. The reviewer is correct to remind us that it is important to ensure a good match between the AN model and cortex neuron BFs, and we therefore re-ran our AN model analyses to include only frequencies in the 0.3-22.8 kHz range that is also covered by the cortical data. We have revised the manuscript text and figures to reflect this.

Line 226-227 – "...very substantially and statistically significantly smaller...", please change to "...significantly smaller..."

Done.

Line 233 & 234 – please change "beat" on both lines to "the musical beat"

Done.

Line 248-249 – The logic behind this statistical test is hard to follow. What is a "consensus OOR"? This wasn't defined earlier. Was the pool of possible OORs created for each unit and then pooled together, or was it created based on the summed firing rates across units? Lastly, using a sign test seems inappropriate here, since in Fig 2C the distribution of OORs for auditory cortical neurons is definitely bimodal. I prefer something akin to lines 255-257, where the number of OORs above the pool OORs is computed.

We agree with the reviewer that the original wording was unclear. The pool of possible OORs was calculated on the population firing rate (i.e. firing rate of all multiunits averaged together). We also no longer use the sign test and instead report the interquartile range of the distributions and the number of songs with a beat contrast above the 95th percentile. This paragraph now reads:

The pool of possible BCs based on the population cortical firing rate is shown in Fig. 3A for an example musical excerpt (see Supplementary Fig. S1 for all excerpts), and the distribution of these values is shown in Fig. 3B. Fig. 3C compares the median BC across all beat interpretations for each musical excerpt with that excerpt's BC at the consensus beat perceived by listeners. Cortical BCs at the consensus beat were significantly larger than the median of the pool of possible BCs ($p < 10^{-4}$, Wilcoxon paired sign rank test, $N=20$ songs). Fig. 3D-3F shows BCs based on the AN model (see Supplementary Fig. S2 for all excerpts). AN model BCs at the perceived beat were also significantly larger than the median of all possible BCs (Fig. 3F; $p < 0.001$, Wilcoxon paired sign rank test, $N=20$ songs). However, there are some notable differences between cortical and AN model BCs. Cortical responses resulted in wider range of BC values, as evidenced by the higher interquartile range of the cortical BC distributions compared to AN model BC distributions ($p < 10^{-4}$, Wilcoxon paired signed-rank test, $N=20$ songs). Furthermore, cortical BCs at the consensus beat were above the 95th percentile of all possible BCs for 14 of the 20 musical

excerpts (Supplementary Fig. S3), but for only 10 out of 20 excerpts for AN model responses (Supplementary Fig. S4). Together, these results suggest that a large BC is a feature that distinguishes the consensus beat from most other possible beat interpretations, and that two important consequences of auditory processing may be an amplification of small BCs already present at the auditory periphery, and a further restriction of the candidate beat interpretations that would result in large BCs.

Line 287 – "...varied..." please change to "...correlated..."

Done.

Line 292-293 – "unlikely to be...neural responses", please discuss further why this buildup might occur if it doesn't occur in auditory cortex

We are not claiming that entrainment does not occur in the auditory cortex, only that we don't observe it in our anesthetised spike rate data. The mention of entrainment was only in reference to current neurophysiological models of beat perception from human studies using noninvasive imaging methods. We now clarify this:

...unlikely to be due to cortical entrainment or build-up in neural responses that are typically reported in human studies of musical beat perception.

Line 300 – "...nearly 90% of the variance..." change to "...most of the variance..."

Done.

Line 318-320 – I worry this sentence could be misleading, because without the nonlinearity the linear STRF model assumes a gaussian distribution of firing rates, which is a poor assumption because the firing rates can't be less than zero. If you state the purpose of the nonlinearity in the Methods (to constrain the range of firing rates), I think you can cut this sentence.

Cut this sentence and added the purpose of the nonlinearity to the Methods.

Line 333 – "...these effects...", you should reiterate that this does not include the relationship between tapping consensus and OOR over time (Fig. 4a and b), and discuss this later in the Discussion (see my comment on line 292-293)

Good point. We've added an additional sentence to the discussion highlighting this:

Another aspect of musical beat perception that our cortical data do not explain is the buildup in the perception of beat, which is not accompanied by an increase in beat contrast at the sensory representation level (compare Figs. 4B and 4C). Future work could explore neural correlates for the buildup in musical beat perception, and explore which features in the sensory representation might determine the timecourse for this buildup.

Line 363-365 – "...then part...rhythmic groupings." I don't see how this follows the previous half of the sentence, please change. In particular, I think referring to music composition here is out of context, since earlier in this paragraph you refer specifically to perceptual accents and nothing relating to how the music is constructed.

This sentence was an aside. It is not important. Since it was found to be distracting/unclear, we have removed it.

Line 413 – "...understand..." change to "...to understand..."

Thanks for catching this, done.

--

Reviewer: 2

Comments to the Author(s)

The authors describe an experiment in which firing rates in rat auditory cortex are linked to the metrical structure of rhythms, as found by human tapping preferences. More specifically, they found that 1) firing rates were higher to sounds on than off the beat, 2) the ratio between on-beat and off-beat firing was largest for the beat interpretation that humans preferred, 3) these findings were similar, but much smaller for an auditory nerve model, 4) the ratio between on-beat and off-beat firing correlated with tapping consensus amongst humans, and 5) the on-beat off-beat ratio can be explained by a spectrotemporal tuning model.

The topic of this paper is interesting. The methods used seem solid (though I am not a specialist in animal research, so I cannot comment on the methods used for the recordings in rodents). Mainly, I have some comments on the framing of the research question and results. Also, the methods lack explanation for some of the choices made (see below for specifics).

The main conceptual issue:

1. The authors describe their question as "does neural emphasis in the auditory representation of a rhythm predispose the musical beat?" (p 3, lines 49-51). Throughout the paper, the way this question is framed seems to suggest that this neural emphasis is part of the mechanisms underlying beat perception (for example: "beat perception may begin weakly at the ear"). Also, in the introduction, only theories related to beat perception are discussed. However, as the authors themselves note in the discussion, fluctuations in firing rate are "merely a preprocessing step" (line 375) and "a physiological correlate of perceptual accents" (line 360). To me, it seems that the results presented here have more to do with perceptual accenting than beat perception. The relationship between those two has been examined quite a bit before (for example by Povel and Essens in the '80). What this paper adds is a possible neural mechanism for perceptual accents (higher firing rates to accented than unaccented tones), which is an important contribution. But in my opinion, this should be framed in terms of perceptual accents, already in the introduction. That the results are based on beat locations in humans probably stems from these beat locations being based on (perceptual) accents. Therefore, in a way, the beat location is just a proxy for the location of the perceptual accents. I think the manuscript would improve if this was more clear. To solve this, I think it would be good to move the discussion about perceptual accents partly to the introduction, and explain the relationship between firing rates and perceptual accents, and perceptual accents and the beat more explicitly.

This is a good suggestion which we have tried to accommodate by expanding paragraph 2 of the introduction, introducing the classic work by Fraisse and by Povel and Essens right from the start, and making readers more aware of the context of the interplay between perceptual accents and the perception of regular beats, in which our results need to be interpreted. To be clear though, we are investigating beat perception in this paper, not perceptual accents. The human behavioural data are of people tapping to the musical beat at more or less isochronous intervals. We do not have a similar marker for perceptual accents - this would be an interesting question, but is a different study altogether.

However, we take the reviewer's point that framing our work in the context of perceptual accents from the start would be helpful to the reader. The relevant section of the introduction now reads:

If points of relative neural “emphasis” in the ascending auditory representation of rhythms predispose the induction of musical beat, then this could help explain why people largely agree on when beats occur. A key assumption here is that localised, transient increases in firing rates of auditory neurons would lead to points of perceptual emphasis, and that the temporal structure of these points in turn shapes the perception of a periodic beat. The idea that perceptual emphasis and beat perception are likely to be intimately linked is not new. Pioneering work by Paul Fraisse explored how differences in a sound feature (e.g. intensity, pitch) in a series of isochronous sounds immediately evoke the perception of rhythmic groupings (see e.g. (14) for a review), and Povel and Essens' (15) empirical model of beat perception suggests that the beat aligns itself maximally to "perceptual accents" resulting from changes in a sound feature or temporal context. Nowadays, the cortical activity evoked by music in the human brain is thought to arise from interactions between relatively low-level evoked responses in the incoming sensory stream and “higher-level” or feedback mechanisms that may include the entrainment of cortical oscillations (16). Thus, a clearer understanding of the bottom-up neural signals evoked by music could shed light on how oscillatory dynamics in the brain entrain to auditory rhythms (17–22), as well as on the role played by the motor system in finding and maintaining a regular pulse (23–33).

General issue:

2. The organization of the paper is a bit confusing sometimes. Some information is repeated (for example the first sentence of the results), and sometimes information in the results section refers to an analysis that was not mentioned in the methods section (for example the auditory nerve model). It would help if all methods used were present in the methods section and omitted from the results section.

We've deleted the first sentence of the results, thanks for flagging this. Also, apologies for omitting the AN model description from the original methods section! We have rectified this in the revised manuscript.

Other issues:

3. Line 18: "Beat perception is regarded as a high-level ability". Is it? Later the authors describe entrainment models. Why would entrainment be "high-level"? Do you mean cortical vs. subcortical? Though beat perception has also been linked to the basal ganglia. How is this high-level?

We thought that this would have been a relatively uncontroversial statement. The fact that beat perception is predictive (consider negative beat asynchrony) and not just reactive requires fairly sophisticated processing. Also, many authors seem to think that "proper" beat perception involves not just cortical, but also basal ganglia, circuitry. Premotor cortex - basal ganglia loops are even further removed from auditory inputs than sensory auditory cortex, making them "higher" in a "bottom up" sensory processing chain sense. Thus, our reading of the literature would have led us to think that "beat perception" is "high-level" both in an anatomical and in a figurative sense. So it is not entirely clear to us why the reviewer is taking issue with this statement. However, quite how "high-level" beat perception is, or which way is "up", are not discussions that are central to this paper, and we hope the reviewer will allow us to sidestep this issue by rewording this sentence thus:

Previous research has shown that musical beat perception is a surprisingly complex phenomenon involving widespread neural coordination across higher-order sensory, motor, and cognitive areas.

4. Line 21: *I suggest leaving out the word "surprising", given the previous paper by the same authors showing similar results for artificial stimuli.*

Done.

5. Line 24: *This sentence is unclear (it reads as if 40 humans reported the firing rates)*

Agreed, this sentence was unclear. We've revised it to the following:

Extracellular firing rates in the rat auditory cortex were recorded in response to twenty musical excerpts diverse in tempo and genre, for which musical beat perception had been characterised by the tapping behaviour of 40 human listeners. We found that firing rates in the rat auditory cortex were on average higher on the beat than off the beat.

6. Line 28: *I do not think your findings suggest that bottom-up processing "facilitates the*

extraction of a beat". Bottom-up processing is responsible for created perceptually accented tones, which in turn can induce a beat (not necessarily "facilitate" it).

We've removed the word "facilitate" and reworded this to read:

These findings strongly suggest that the "bottom-up" processing of music performed by the auditory system predisposes the timing and clarity of the perceived musical beat.

7. Line 61: typo (remove "be")

Done.

8. Line 77: *I do not have access to the stimuli from the MIREX competition (the website does not give me permission), so I would appreciate more information (could also be in the supplementary information). Is this instrumental music, what are the metrical structures, did any loudness normalization take place?*

We have uploaded the stimuli along with the data to the Dryad repository. This is stated at the beginning of the Methods section. The reviewer and interested readers can thus access all the stimuli, listen to them or process them as they see fit. The stimuli comprise a diverse and eclectic set of music, from pop to electronic to classical music, and from very slow adagios to fast paced dance music. We do not have access to the corresponding sheet music, so we cannot exactly specify the time signatures or tempos the composers intended. However, we have added a table to the supplementary information with some basic information about each excerpt, including title, artist, genre, and tempo (as determined by our analysis of the 40 human listeners tapping to the beat - copied below). Other than scaling each stimulus so as to present them all at a standard mean sound level of approximately 80 dB SPL and a resampling to the non-standard 48,828 Hz sample rate of our TDT stimulus delivery system, no other manipulation or loudness equalisation of the sound waveforms was performed.

Excerpt	Title	Artist	Genre	BPM
1	You're The First, The Last, My Everything	Barry White	R&B/Soul	129
2	A New England	Billy Bragg	Alternative/Indie/Folk	82

3	El Contrapunto	Los Mensajeros De Las Libertad	Latin/Folk/World	152
4	Green Eyes	Erykah Badu	Contemporary Soul	42
5	Passe & Medio Den Iersten Gaillar	Josquin Des Prez Thai China Dolls	Classical	68
6	Wo Ai Ni		Pop	82
7	Le Sacre du Printemps	Igor Stravinsky	Classical	57
8	Flim	Aphex Twin	Electronica	148
9	Hurricane	Bob Dylan	Classic Rock	127
10	Vespro della beata Vergine	Claudio Monteverdi	Classical/Opera	62
11	Komm nach Tirol	Zillertaler Schürzenjager	Folk	141
12	Matthäus-Passion, BWV 244	Johann Sebastian Bach	Classical	54
13	Kalasnjikov	Goran Bregović	Rock/Jazz	181
14	Not Gonna Get Us	t.A.T.u.	Pop	130
15	Le Bruit Du Frigo	Mano Negra	Rock/Folk	63
16	La Carpinese (Tarantella)	Lucilla Galeazzi	Classical	92
17	The Piano Has Been Drinking [Not Me]	Tom Waits	Alternative/Indie	46
18	Exit Music (For A Film)	Radiohead	Alternative Rock	62
19	Possessed To Skate	Suicidal Tendencies	Metal	190
20	El Gato Lopez	Ska-P	Ska	222

9. Related to this: Is there any way to link the results of the firing rates to the locations of accented vs. unaccented tones (instead of the beat as derived from the tapping data)

in this dataset? For example, could you maybe provide an average waveform for the sound on the beat and the sound off the beat? Looking at Figure 1 it seems that there may have been differences in loudness or loudness-changes (onsets) for beat vs. offbeat sounds. This could very well explain your data, so it would be interesting to see whether this is the case (e.g., whether here, the accents may not just have been perceptual, but also in loudness).

This is an interesting question, but again we are not aware of a computational method to determine whether events in music are perceptually “accented or unaccented.” On the question of loudness, however, the AN model data that we have included in our manuscript are probably the closest possible thing that one could provide to give an “average waveform on the beat”. The reviewer needs to bear in mind that the raw waveforms will be dominated by relatively high frequencies which do not have a precisely constant phase from one beat to the next, and which will therefore destructively interfere and cancel in any such beat triggered averaging of the raw waveform. The auditory nerve model we used will overcome this problem by performing a kind of envelope extraction and/or half-wave rectification for relevant frequency bands, and auditory nerve fiber firing rates are monotonically related to sound intensity (loudness). We therefore invite the reviewer to consider Figure 2E as our attempt to address exactly the point raised here, only that we are looking at the stimuli through a lens that captures the physiologically relevant preprocessing provided by the auditory periphery.

10. Line 103: Could you offer some explanation as to why you chose this way of estimating the beat location from the tapping data? Why not the average time? Also, you chose the beat in such a way that beat period was constant. But if I am understanding correctly, you used real music. How do you then account for possible tempo changes?

The reviewer’s question “Why not the average time?” we don’t quite understand. Our choice of pooling, smoothing, and using a peak finder seemed like a reasonable and straightforward way of finding a consensus beat interpretation, and in all cases this method gave us a clear answer that we checked visually afterwards and found to be agreeable. Supplementary Fig. S5 also gives the reviewer and reader the opportunity to see whether beat locations are in agreement with the tapping data. Perhaps there was a better way to estimate beat locations but it would have helped to have a more specific suggestion from the reviewer.

Our musical excerpts were only 10 seconds long, and none of the excerpts contained abrupt tempo changes during this time. It is true that real music may not have a strictly

isochronous beat. However, inspection of the tapping behaviour (shown for all listeners and all songs in supplementary figure S5) does not suggest that the listeners typically tried to do anything other than find a set of isochronous intervals that matched their perception of the music as they tapped along to the beat. Our method for estimating beat location is designed to estimate the most likely consensus time points for the start and end of these isochronous intervals. We are definitely open to the suggestion that there may be other, perhaps better methods of estimating beat locations, but note that our analysis of the neural data used fairly broad "ON" and "OFF" response windows and would therefore not be very sensitive to inaccuracies on the order of a few milliseconds in the estimation of beat locations. Bearing this insensitivity to precise beat locations in mind, we don't think there is much to be gained by evaluating alternative methods for estimating beat locations from the tapping data.

11. Line 114: I am a bit surprised by the beat ranges. 0.7 and 3.7 Hz seem well outside of the rate at which humans normally can perceive a beat. How do you explain this? Could it be that the tapping was at subdivisions or higher-level meter instead of the beat? How do you account for this in the data?

Beat perception at its extremes ranges from 0.5 – 4 Hz (London J. 2012 Hearing in time. Oxford, UK: Oxford University Press), and the tapping data here are not outside of this range. It is also worth bearing in mind that the set of musical excerpts used in the MIREX database was deliberately chosen by the MIREX team to provide a challenging and diverse test set for artificial beat finding algorithms, and it may therefore by design cover an deliberately wide range of tempos and genres. Other than that we don't really have an explanation to offer why the listeners tapped at the intervals that they tapped at. The data are what they are, and informally, when we listened to the excerpts and compared our own perceptions against the recorded tap times (which are all in supplementary figure S5) we found nothing unusual or suspicious or implausible in the tapping data. It is certainly the case that for several songs, some listeners would tap at two times or even three times the rate of others, suggesting that the tapping at subdivisions of the meter that the reviewer surmises does frequently occur. However, for the very fastest songs (excerpts 19 and 20) there was a remarkably high level of consensus among the listeners, and given the tapping data, for those two songs it seems impossible to escape the conclusion that the beats were perceived at a rate in excess of 3 Hz by the very large majority of listeners.

12. Line 141: Why did you use this metric for tapping consensus? Why not just the spread in tapping times?

Reviewer 1 raised a similar question in his/her comment captioned "Line 152-155" and you can see further clarification there, but briefly, a metric based on the spread (normally quantified as mean square error) would disproportionately penalise tapping behaviour where a minority of listeners perceive a beat that is precisely twice as fast but at an identical phase of that heard by a majority of listeners. Every other tap would incur either a near-zero error or a maximal half beat interval squared error. Our metric is thus designed to not be oversensitive to this type of "error". We now mention this in the revised text.

13. Line 157: Was data averaged over the three rats? Probably, but maybe mention this somewhere.

Average response data were indeed averaged across all animals. We now mention this explicitly in the first sentence of the revised Results section.

14. Line 222: The rationale for the auditory nerve model only becomes clear later in the paper. This could be added to the methods, and maybe even the introduction, to explain that you will compare your results to this model, and why.

This is a good suggestion which also relates also to this reviewer's "general point 2". As mentioned earlier, we have added a paragraph to the revised Methods to describe the rationale for the AN modelling and the methodology. As suggested here, we now also mention the AN model in the revised introduction.

15. Line 249: typo (Figure 3C is now 3D?)

Fixed - thanks!

16. Line 365-368: This sentence was unclear to me, and it does not seem to relate to the previous.

We have removed this sentence.

17. Line 370: "Interaction between perceptual accents and temporal structure" is a bit unclear. Do you maybe mean the "temporal structure of the accents"?

We have elaborated this part a bit to make it clearer. We believe that the features that cortical neurons are particularly sensitive to a periods of high temporal contrast, and that such events are therefore “accented” in cortical responses. Spelling this out should make it more obvious how neural mechanisms that set perceptual accents might interact with the temporal structure of music more obvious. The relevant section now reads:

Our work suggests that transient increases in firing rate might be a physiological correlate of perceptual accents, since adaptation would predict larger neural responses for stimulus features that have not been experienced in the recent past. We think it is likely that the combination of excitatory and inhibitory receptive field elements in close succession, which is frequently observed in cortical neurons and well captured by LN models, heightens the sensitivity of cortical neurons to temporal contrast. Thus, our data support the notions that auditory processing gives rise to perceptual accents, for example through the detection of high temporal contrast events, and that musical beat perception arises from an interaction between perceptual accents and the temporal structure of music...

18. Line 416: Beat perception is not an ability that only humans possess. Several species have been shown to be capable of beat perception (Snowball, see Patel et al., 2009; Ronan, see Cook et al., 2013), and recently, Hugo Merchant’s lab also showed that rhesus monkeys, with the right incentive, can be taught to tap to a metronome, see Gamez et al., 2018. While you could question whether the latter is really beat perception, it would be good to nuance the statement in this sentence a bit.

Fair point. This sentence now reads:

Finally, if musical beat perception is a rare and unusual ability that only humans and a few select species may possess, then it invites the question of where, functionally and anatomically, we diverged from other species to be able to perceive musical beat.

We also preface this sentence in the previous paragraph by citing the suggested papers:

For example, it has been recently shown that non-human primates (76,77) can predictively synchronise to metronomes. Additionally, some songbird (78) and pinniped (79) species can also find and synchronise to the beat in music.

Appendix B

3 February 2020

To the reviewers:

Thank you again for your time and effort in reviewing our revised work. Below we address your final recommendations one by one.

Sincerely,

Vani, Nicol, Jan

Reviewer: 1

Comments to the Author(s)

The edits the authors have made have considerably improved the manuscript. However, I spotted several more minor issues that I think should be corrected:

Line 27 -- "interpretations of beat" change to "interpretations of the beat"

Fixed.

Line 41 -- "despite musical beat being", maybe "despite musical beats being" or "despite the musical beat being"

Changed to “despite musical beats being...” so sentence now reads:

Secondly, despite musical beats being a subjective percept rather than an acoustic feature of music (5), individual listeners tend to overwhelmingly agree on when beats occur.

Methods -- Generally the flow of the methods section is much better than in the previous version. However, the "On-beat, and off-beat, and beat contrast (BC)" section is out of place because it refers to analyses of neural activity before the method of collecting and preprocessing the neural activity is described. This section should go after

preprocessing. Also, I think the title of the section could be changed to "neural data analysis", or at the very least please remove the first "and" in the title.

Agreed. This section now comes after the “Data preprocessing” section as suggested, and we also removed the extra “and” from the title.

Line 167 -- "...but are e.g. twice the speed..." Please be more direct rather than using "e.g." "...a multiple of the speed..." or "...two or three times the speed..." could work, whichever is more accurate.

Good suggestions, changed this to “... a multiple of the speed...”.

Line 232 -- "in figures...", are you referring to Fig S7?

Yes, thanks for catching this omission, fixed now.

Line 340-341 -- "...are unlikely to be due...human studies of musical beat perception." Please provide a citation.

Done.

Line 360-362 -- Thank you for including the STRFs for all of the units in Fig. S7 and their beat contrasts, it is very informative. However, I think you should also include the plot you provided in the reviewer comments of beat contrast vs CF as another supplemental figure, because the relationship (or specifically the lack of a relationship) between beat contrast and CF can be hard to decipher from S7 alone, and it also shows the distribution of CFs of the population. When you include that figure, I recommend making the dots bigger so they are easier to see.

Added the beat contrast vs. CF plot as supplementary fig. S8 (see below) and also refer to it in the manuscript.

Fig. 5a -- please include a colorbar, something like what you did for Fig. S7 is sufficient. Yes, thanks for spotting this. Below is the updated Fig. 5.

Line 422-423 -- "...do not only occur periodically at the beat (see Fig. 1E)." This is a bit hard to decipher, do you mean that increases in activity can occur off the beat as well?

This sentence was only meant to emphasise that the large responses we observed in the auditory cortex were explained by spectrotemporal contrast sensitivity, and thus are not specific to musical beats. We agree that this sentence was not very clear. The start of that paragraph now reads:

It is important to emphasise that the large responses we observed in the auditory cortex occur at points of high spectrotemporal contrast in the sound and do not

selectively occur on musical beats (see Fig. 1E for examples of large responses that do and do not coincide with musical beats). These large fluctuations in firing rate therefore represent a pre-processing step that, we believe, constrains periodic activity at subsequent stages of processing where a single periodic interpretation of beat is selected.

Line 458 -- "...perceiving musical beat" change to "perceiving a musical beat".

Done – thanks for these suggestions!

Reviewer: 2

Comments to the Author(s)

The authors have addressed all my comments, and the paper has been improved substantially.

Thank you!